# AI-NERD: Elucidation of relaxation dynamics beyond equilibrium through AI-informed X-ray photon correlation spectroscopy

James P. Horwath [1] ✉, Xiao-Min Lin [2], Hongrui He[3,4], Qingteng Zhang[1], Eric M. Dufresne[1], Miaoqi Chu[1], Subramanian K.R.S. Sankaranarayanan[2,5], Wei Chen [3,4], Suresh Narayanan[1] ✉ & Mathew J. Cherukara [1] ✉

Understanding and interpreting dynamics of functional materials in situ is a grand challenge in physics and materials science due to the difficulty of experimentally probing materials at varied length and time scales. X-ray photon correlation spectroscopy (XPCS) is uniquely well-suited for characterizing materials dynamics over wide-ranging time scales. However, spatial and temporal heterogeneity in material behavior can make interpretation of experimental XPCS data difficult. In this work, we have developed an unsupervised deep learning (DL) framework for automated classification of relaxation dynamics from experimental data without requiring any prior physical knowledge of the system. We demonstrate how this method can be used to accelerate exploration of large datasets to identify samples of interest, and we apply this approach to directly correlate microscopic dynamics with macroscopic properties of a model system. Importantly, this DL framework is material and process agnostic, marking a concrete step towards autonomous materials discovery.

Structure-property relationships are the core of materials science and condensed matter physics. However, defects and disorders make it difficult to describe real materials with simple analytical models. Adding to the complication, many materials are out of equilibrium and either evolve through time or are meta-stable, meaning their structure and properties can experience significant and fundamental changes due to small changes in the environment. In these cases, structural and microstructural analysis are not enough to understand the system, and the dynamics of the system must be studied to understand how the temporal evolution of the microstructure manifests in bulk materials properties.

Synchrotron X-ray scattering is ideally suited to studying structure and dynamics in complex systems due to high illumination brightness, flux, and coherence combined with state-of-the-art X-ray detectors and operando capabilities[1–8]. In particular, x-ray photon correlation spectroscopy (XPCS) is capable of capturing material dynamics with time resolution spanning μs - hours and spatial resolution ranging from sub-nm - μm. Flexible experimental conditions make XPCS experiments compatible with a variety of operando environments. In this work, we consider the case of rheo-XPCS, wherein the partially coherent X-ray beam illuminates an X-ray transparent rheometer stage which can simultaneously shear the sample and measure mechanical response. Built on the same fundamental mechanisms as Dynamic light scattering (DLS), XPCS measures dynamics via temporal decorrelation of scattered X-ray intensities[9–11]. A schematic of XPCS is shown in Fig. 1A and B. A coherent X-ray beam scatters off of a sample and produces a scattering pattern on a pixel-array detector in the far field. The intensity autocorrelation is

[1]Advanced Photon Source, Argonne National Laboratory, Lemont, IL, USA. [2]Center for Nanoscale Materials, Argonne National Laboratory, Lemont, IL, USA. [3]Materials Science Division and Center for Molecular Engineering, Argonne National Laboratory, Lemont, IL, USA. [4]Pritzker School of Molecular Engineering, University of Chicago, Chicago, IL, USA. [5]Department of Mechanical and Industrial Engineering, University of Illinois, Chicago, IL, USA. ✉e-mail: jhorwath@anl.gov; sureshn@anl.gov; mcherukara@anl.gov

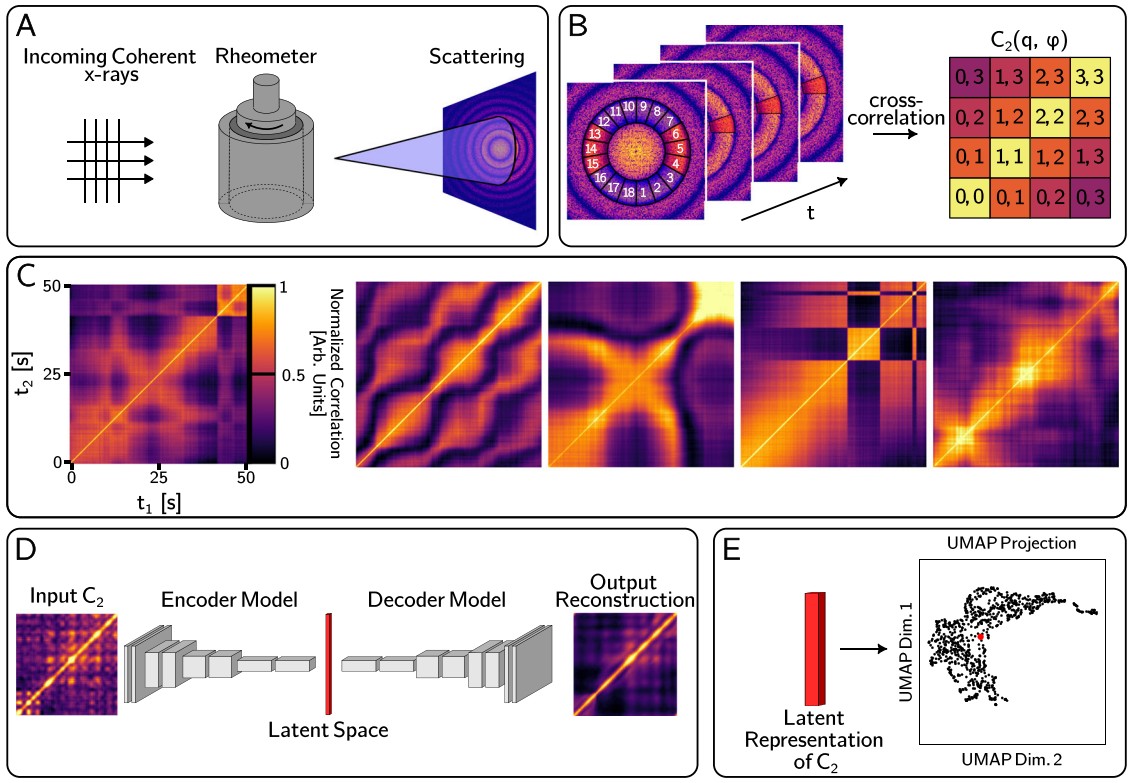

**Fig. 1 | Schematic of the experimental setup and machine learning workflow for x-ray photon correlation spectroscopy (XPCS) data. A** In rheo-XPCS, a rheometer is placed in the beam path so that coherent X-rays scatter off the relaxing sample. **B** XPCS two-time correlations, $C_2(q, \phi)$ are calculated by correlating intensity in a specific scattering region over time experimental times $t_1$ and $t_2$, where $q$ represents the radial scattering wavenumber and $\phi$ describes the azimuthal scattering bin over which $C_2$ is calculated. **C** shows a sample of experimental $C_2$ to illustrate the wide variation in dynamics seen in non-equilibrium XPCS. The time scale bar in the left-most $C_2$ applies to all other images. **D** The autoencoder is trained to reproduce raw $C_2$, and the learned latent representation is used to cluster and classify data points (**E**).

calculated between all pairs scattering frames, collected at times $t_1$ and $t_2$, at a fixed scattering vector, **Q**, and is plotted as a two-time correlation function, called $C_2(\mathbf{Q}, t_1, t_2)$. XPCS correlations are calculated in groups at several **Q** from the same raw scattering data; in Fig. 1B, we illustrate that, for the data shown in this work, the scattering pattern is divided into 18 azimuthal bins along the structure factor peak so that $C_2$ from each bin can be compared to gauge the isotropy of dynamics. For systems at steady-state, where dynamics do not change with time, a rigorous framework for extracting physical properties from XPCS data exists. In these cases, intensity is averaged over equivalent delay times, $\tau = t_1 - t_2$, to produce a one-dimensional plot of the time-averaged autocorrelation function, $g_2(q, \tau)$, which is fit to well-established analytical or empirical models to extract physical information. In contrast to $g_2$ which averages over $\tau$ and therefore cannot express non-equilibrium dynamics, $C_2$ is capable of describing any type of relaxation dynamics and provides a "fingerprint" of the non-equilibrium system at any given experimental time. A variety of analyses have been used to take advantage of both the fundamental foundation for traditional XPCS analysis and the information-rich correlations which describe changes between specific time points[12–15,15–18], however, the amount of human adjudication required for interpretation of results from such advanced XPCS analysis methods, as well as the amount of data collected in synchrotron experiments, pose significant barriers to the development of a more quantitative physical understanding of dynamics in complex material systems. To further complicate the matter, the variety of patterns shown in experimental $C_2$ from a single system varies drastically such that even visual identification of relationships between data points is difficult (see Fig. 1 C and Supplemental Fig. 1 for a sample of $C_2$ data). The limitation imposed by data interpretation bandwidth will become even more pronounced with the use of high-frame-rate, large-pixel-array x-ray detectors and the world-wide commissioning of ultra-brilliant fourth-generation synchrotron facilities[19–26].

Recent years have seen a tremendous increase in the application of machine learning (ML) methods to scientific data with applications ranging from assisting medical diagnosis and guiding autonomous vehicles, to solving fundamental physical problems[27–29]. Specific to x-ray characterization, ML methods are being used across nearly every characterization technique[30]. Examples include the use of ML to determine the structure-property relationship[31–35], accelerate and enhance coherent characterization techniques[36–41], accelerate emission spectroscopy, reduce dose and noise in tomography, and accelerate Bragg peak fitting[42–45]. In the XPCS community, recent work has demonstrated the use of ML to denoise $C_2$, which lead to significant improvement of the quantitative interpretation of the XPCS results and detection of anomalous results, and using ML to link physical parameters with $C_2$ topology from simulations to further our understanding of the origin of complexities in XPCS data[46–48].

Here, we use rheo-XPCS studies of relaxation in a glassy colloidal suspension as an example system to develop a material- and process-agnostic AI toolbox for exploring and understanding non-equilibrium XPCS $C_2$. Dynamic heterogeneity and non-linear rheological response, both resulting from highly heterogeneous distributions of constituent particles and their local cooperative motion, are well known in glassy systems and lead to the observation of highly non-equilibrium relaxation dynamics[49]. Understanding the role of microstructure, jamming, and local heterogeneities on macroscopic properties is an active area of research in complex fluids, granular materials, and other fields studying non-equilibrium dynamics[50,51]. Among very few experimental techniques that can study this heterogeneity, XPCS has

the advantages of providing information with high spatial and temporal resolution[52], and the ability to probe structural and mechanical response simultaneously through the use of the operando rheological environment.

We present our unsupervised framework for the exploration of large, complex XPCS datasets, called AI-NERD - Artificial Intelligence for Non-Equilibrium Relaxation Dynamics. We demonstrate the development of a convolutional autoencoder (AE) for encoding $C_2$ into a reduced space and then apply K-Means Clustering to classify data points based on their position in the latent space. Next, we exhibit the utility of this type of analysis in a representative case, namely tracking non-equilibrium relaxation dynamics in a colloidal glass. We illustrate how classifying $C_2$, and how comparing transitions between classes as a function of time can map to rheological measurements showing the evolution of shear stress within the material. Finally, we present how our method can be used to take in user-specified $C_2$ of interest and return other samples from the dataset in order of similarity. In comparison to other applications of representation learning to understand materials systems[47,48,53,54], our method is the first to use unsupervised learning in time-resolved experiments to guide understanding of non-equilibrium dynamics without requiring background information on the sample or experimental setup. Rapid changes in non-equilibrium dynamics are invaluable for understanding how materials respond to stimuli in situ across wide ranges in space and time, however capturing these fluctuations is difficult with experiments outside of time-resolved coherent X-ray scattering techniques such as XPCS. Increased x-ray flux and coherence at next-generation light sources will increase signal in XPCS correlation calculations and push analysis towards the richer $C_2$ representation of dynamics compared to the traditional $g_2$. Therefore, the need for reliable and general methods for analyzing complex $C_2$ is greater than ever. Our work demonstrates the capability of applied machine learning to accelerate scientific discovery through X-ray characterization and to move towards the ability to fully utilize experimental capabilities for high-frame rate, high-flux/coherence experiments available at next-generation synchrotron light sources.

## Results

### Unsupervised deep learning to elucidate relaxation dynamics

Machine learning models generally can fit into either the supervised, or unsupervised learning paradigms. In supervised learning, scientists provide a labeled dataset which is used to optimize the model weights based on the difference between model predictions and the provided ground truth. Unsupervised learning is used in cases where labeled data is unavailable or difficult to produce, and algorithms generally aim to distill features of the raw data, identify statistical trends across the dataset, or cluster the dataset based on the properties of the data and similarities between data points. Unsupervised learning presents opportunities for reaping the pattern recognition and processing acceleration benefits of machine learning without requiring labeled data or even a physical understanding of the system[55,56]. This is incredibly useful for understating structure dynamics from experimental $C_2$ since non-equilibrium dynamics come in a variety of flavors and are often poorly understood.

Autoencoder architectures perform unsupervised learning by taking raw (or featurized) data as input, passing this through an encoder model that compresses data into a latent representation, and then decoding the latent representation back into the original data dimension. While model weights are optimized by comparing the output reconstruction with the raw input data, the ability of the model to decode the heavily compressed latent representation into an accurate rendering of the input data signifies that the latent space represents a minimal set of fundamental features that can describe the data. In this type of representation learning, ensuing analysis of the distribution of data in the latent space enables the identification of global trends in the data[57,58]. Recent research has applied this approach to one-dimensional spectroscopic datasets, and latent space analysis has been used to relate subtle changes in peak position and shape directly to the physical properties of a material[53,54]. For higher-dimensional data such as XPCS $C_2$, where data can be represented as images, convolutional neural networks (CNN) are able to accurately encode spatial information, and take advantage of the expressive power of deep learning to provide an accurate and adaptive understanding of scientific data[59]. In this type of neural network, sliding window filters act on an image, and spatially resolved weights are learned. Convolutional autoencoders have been demonstrated both in computer vision tasks and as flexible image compression algorithms compatible with the representation learning framework mentioned above[60–62]. We adapt this approach to encode experimental $C_2$ and classify data based on their latent representation. A schematic of our autoencoder and latent space analysis is provided in Fig. 1 D and E.

The development of autoencoder models that enforce continuity and orthogonality in the latent space, such that latent dimensions are independent and hopefully interpretable, is a major concern in the field of representation learning. A common approach for latent space conditioning is seen in variational autoencoders (VAE), where an additional loss term is used in the training process to enforce that latent variables are drawn from multivariate Gaussian distributions[63]. Enforcing the shape of latent variable distributions helps to develop continuity in the latent space. Further loss or regularization terms can be added to the neural network model to enforce orthogonality between latent parameters. These additional constraints produce learned representations which are often more directly interpretable, however, the model training process is significantly more difficult and optimizing these conditioning factors comes at the cost of sacrificing image/data reconstruction quality. For instance, a $\beta$-Autoencoder attempts to address this concern by using a parameter $\beta$ to weight the relative importance of latent space conditioning and reconstruction error, however choice of $\beta$ can be difficult and depends on the exact goal of the ML task[54]. While these approaches have proven successful in the X-ray characterization papers mentioned above, we have found that using a VAE framework drastically deteriorated the quality of output reconstructions such that we could not trust that latent representations corresponded with input data. We attribute the model's inability to accurately represent our data in the presence of latent space training constraints to the large variability in our training set which may make distillation of the data into a fundamental set of parameters difficult. Therefore, we have chosen to use a standard convolutional autoencoder to maximize the amount of information our model can learn at the expense of the guarantee of a continuous latent space.

We have employed an hourglass-style convolutional autoencoder which compresses 256 x 256 pixel $C_2$ into a 64-dimensional latent space and then decodes the latent representation to reproduce the input data. Further details of model optimization and data augmentation are presented in Section IV D. The sample reconstruction shown in Fig. 1D appears as a smoothed version of the raw input. More examples of experimental $C_2$ and corresponding AE outputs can be seen in Supplemental Fig. 1. These results signify accurate model performance - random fluctuations in $C_2$ will be difficult to capture with image filters optimized for performance on an entire dataset, so the absence of high-frequency variation suggests that learned filters focus on more important image features. The output reconstruction from our optimized architecture maintained long-range features and time-scale information such as the position of changes in the width of the diagonal correlation band, and off-diagonal patterns. To ensure that the latent space encoding accurately represents the distribution of experimental data, after training we used the model to generate artificial $C_2$ by adding noise to the latent representation of real $C_2$ and passing the noisy representation through the AI-NERD decoder. In the

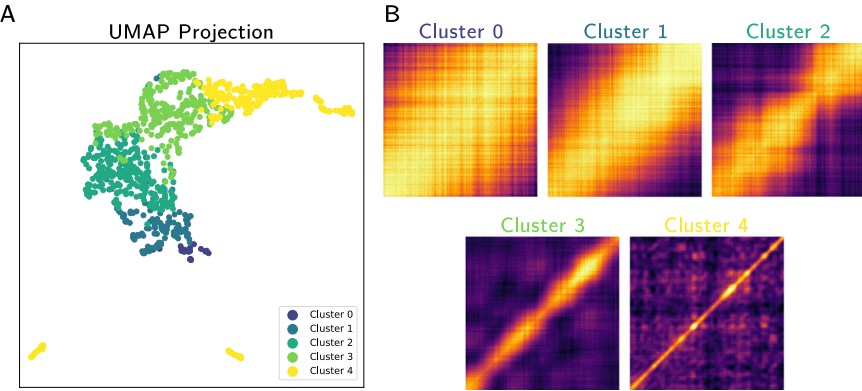

**Fig. 2 | Depiction of the latent space after applying Uniform Manifold Approximation and Projection (UMAP) and k-means clustering.** **A** shows the UMAP visualization of the 64-dimensional latent space. Point colors correspond to the cluster labels defined by k-means clustering with 5 clusters, and match the image labels in (**B**). **B** shows randomly sampled two-time correlations ($C_2$) from each cluster. As high correlation intensity and long-ranging intensity bands show slowly evolving dynamics, we see that increasing cluster number corresponds to increasingly fast relaxation dynamics.

ideal case, passing these artificial latent vectors through the decoder model will produce new data that is similar to the original $C_2$. Supplemental Fig. 2 demonstrates that our model can generate realistic data at a variety of noise levels, suggesting that the latent space is representative of the data. Here, 16 unique artificial $C_2$ were generated by adding Gaussian noise with $\sigma = 0.3$ to the target latent vector. $\sigma = 0.3$ represents the average standard deviation of all latent variables for the encoded test dataset. While generating synthetic data from latent representations is an effective way to ensure that the latent space properly captures the distribution of real data, our goal is not to use AI-NERD as a generative model but rather to explore the distribution of experimental data in a reduced space. Therefore, the moderate noise condition of $\sigma = 0.3$ is sufficient to verify that small regions in the latent space contain similar real data.

### $C_2$ Clustering and latent space analysis

After training the optimized model, a new dataset, corresponding to Rheo-XPCS measurement from a single rheological shear cycle was fed through the encoder model. Unsupervised classification of the new $C_2$ dataset was performed by applying the k-means clustering algorithm directly to the latent representation of the data, determining the ideal number of clusters using the elbow method (Supplemental Fig. 3). This showed that four-six clusters was the ideal number, with fewer clusters separating data solely based on image intensity and more clusters separating the data into unrealistically small groupings.

Due to the high dimensionality of even the latent representation, further embedding is required to visualize the distribution of the encoded data and the clustering results. We used Uniform Manifold Approximation and Projection (UMAP) to transform the latent space of the dataset into two dimensions; this visualization is shown in Fig. 2[64]. UMAP is closely related to t-distributed Stochastic Neighbor Embedding (tSNE), a more common method for non-linear dimensionality reduction[65]. Both of these methods consider the local structure of the data distribution and attempt to project data points onto a lower dimensional manifold, however, in comparison to tSNE, UMAP distorts the data distribution such that it is uniformly distributed in the projection space. This helps maintain the global structure of the dataset and generates projections that are more stable against variation in initialization and hyperparameters than those generated by tSNE[66]. This visualization allows us to qualitatively check the accuracy of the clustering results by seeing whether optimal cluster centers coincide with the densest regions of the UMAP embedding.

Viewing images from each cluster (Fig. 2) shows that relaxation times decrease with increasing cluster labels. Following the trajectory across the UMAP distribution in Fig. 2A from Cluster 0 to Cluster 4, we can see the transition between nearly stationary dynamics in Cluster 0 (high correlation across long times relates to slow structural changes) to slow evolution in Cluster 1 (seen as flat $C_2$ features with lower intensity), to increasingly fast evolution in Clusters 2, 3, and 4. Distances in these embedding spaces should not be quantitatively compared and only serve a qualitative metric of similarity between data points[67]. Outlying yellow regions assigned to Cluster 4 in Fig. 2A appear very similar to the sample image from Cluster 4 in Fig. 2B - they have rapid decay and very low intensity away from the main diagonal. Finally, we note the nearly continuous transition between clusters in the latent space which is evidence that the data set contains a continuous spectrum of relaxation dynamics.

### Probing non-equilibrium dynamics using AI-NERD

With a trained autoencoder and the ability to rapidly encode and classify experimental data in hand, we now describe how this approach can be used to track changing dynamics, explore patterns in large datasets, and guide the selection of quantitative physical models to describe complex systems.

Bringing us one step closer to our goal of bridging information across length scales, our first test aims to understand how fluctuations in rheological measurements correspond with the evolution of the structure and local dynamics of a complex fluid. Full experimental details can be found in Section IV A, briefly, after forcing the system to flow under a high shear rate, the applied shear is removed and the material is allowed to relax to study the impact of non-equilibrium deformation on flow in the complex fluid; simultaneous rheological and X-ray scattering measurements are collected during the relaxation process to track the mechanical response and structural changes, respectively. The rheological data describing relaxation (Fig. 3A) clearly shows non-monotonic behavior, however since the complex $C_2$ patterns seen in experimental data preclude direct quantification of the dynamics, it remains impossible to link these macroscopic changes to microscopic dynamics. To address this, Fig. 3B shows a time-resolved histogram of cluster labels describing the progression of average dynamics through time. The histogram is built by clustering $C_2$ collected from each azimuthal bin (Fig. 1B) at each time step and tabulating the results as vertical, color-coded bars. Vertical black lines in both panels show the times of shear stress minima. The growing peaks of slow dynamics (low cluster numbers) seen between shear stress minima indicate that tracking material behavior through latent space encoding is able to tie intermittent microscopic dynamics to macroscopic rheological changes. Dark peaks seen in Fig. 3B correspond with increases in shear stress and are also seen at later times ($t = 2000$ and 2500 s) where we see no changes in the bulk shear stress

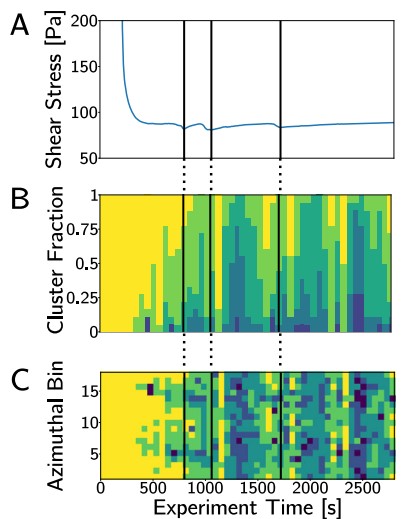

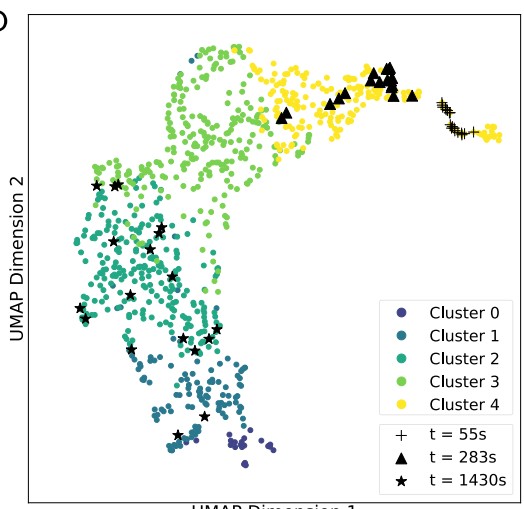

**Fig. 3 | Plotting the cluster distribution of the data as a function of time and azimuthal bin reveals information about the system.** Mechanical response during relaxation is plotted in (**A**) where vertical black lines correspond to shear stress minima (corresponding times are also marked in (**B**) and (**C**)). **B** Shows a time-resolved histogram of cluster labels, where clear peaks of low cluster numbers appear in between shear stress minima. Plotting cluster label as a function of time and azimuthal angle in (**C**) shows clear anisotropy, with slowing dynamics initiated in the flow direction (bins 5 and 14). In (**D**), we show the distribution of data in the latent space (after Uniform Manifold Approximation and Projection, UMAP, embedding) as a function of time to understand how the appearance of $C_2$ change through time, and across scattering angles within a single time step. Note that times in (**D**) do not correspond with the vertical black lines in (**A**), (**B**), and (**C**); the time stamps in (**D**) were chosen to demonstrate that early in the relaxation process data are tightly clustered with no angular dependence, and that as the process continues the dynamics spread across clusters and develop strong anisotropic characteristics.

measurement. Since the incident X-rays probe local dynamics while rheology captures the average behavior of the macroscopic system, the presence of these peaks highlights that XPCS dynamics which happen within the scattering volume of the incident X-ray beam but may not manifest in the macroscopic response. Many physical systems ranging from granular materials in industrial systems to geological motion and earthquakes depend on intermittent, avalanche-like dynamics which may not be present in measurements of the average properties of the system. Thus, the impact of data-driven methods on dynamics which link anomalies with their physical response across length and time-scales is far reaching[68–70].

Analysis of cluster labels as a function of azimuthal scattering angle provides more information about the system. Figure 3C displays a heatmap of cluster labels as a function of the azimuthal bin (Fig. 1B) and experimental time. We observe that dynamics are anisotropic, highlighting the presence of directed motion. Through the cluster labels, we see that dynamics initially slow down in the flow direction (parallel to the applied shear, bins 5 and 14) with comparatively faster dynamics in the vorticity direction perpendicular to the applied shear (bins 1, 9, 10, and 18). Finally, in Fig. 3D we observe the distribution of dynamics throughout the experiment by selecting three experimental time points, and projecting all 18 azimuthal $C_2$ associated with each time point into the latent dimension UMAP plot. We see that, in agreement with Fig. 3C, the initial stages show tight groups of similar dynamics for all $C_2$ collected at a given time, while as time progress the distribution in the latent space spreads drastically. Since our ML model learns to encode spatial features of $C_2$ into the latent space, the position and spread of latent space points correspond to the appearance of $C_2$. The wide latent space distribution from points collected at later times shows that, in addition to the angular dependence of relaxation rate, the visual appearance of $C_2$ differs significantly with the scattering angle.

While XPCS is capable of precisely capturing material dynamics by measuring the decorrelation time between successive scattering frames, a physical model of the process is still required to extract quantitative information describing, for example, particle motion or relaxation rate in the sample. Model selection is difficult for non-

equilibrium systems, where either physical models do not exist, or the selection of a model is complicated by the presence of rapid unexpected changes in dynamics. Furthermore, though theoretical descriptions of dynamics often address the mean behavior of a system, XPCS experiments probe local dynamics which may differ from expected behavior on experimental time scales. Selection of a model based on visual inspection of a few sample $C_2$ from an entire experiment is difficult and potentially unreliable. Therefore, the ability to categorize different types of XPCS $C_2$ patterns to guide the development of structural and dynamics models for evolving materials is crucial.

Results from analysis in Fig. 3A–D suggest that: 1. Dynamics slow down through time, initiated in the flow direction, 2. the intermittent microscopic dynamics are tied to macroscopic rheological fluctuations (however, not all microstructural changes are reflected in the macroscopic response), and 3. in addition to the rate of dynamics, the appearance of $C_2$ differs drastically depending on scattering angle. These conclusions guide us towards the selection of the heterodyne scattering model to describe our data, which occurs when a difference in velocity between two or more dynamically distinct components in the system leads to constructive/destructive interference in correlation patterns and enables the extraction of system-independent information about the relative velocity between components[71–73]. Heterodyne scattering produces a uniquely recognizable fringe pattern in XPCS $C_2$ in specific scattering directions. This phenomenon has been documented in some experimental systems, however, it has not been observed in experiments with colloidal glasses. Thus, our AI-guided model selection provides unique insight into microstructure evolution in complex materials and enables future research to quantify the properties and dynamics in this system.

Supplemental Fig. 4 shows the UMAP latent space, where each pixel is colored by the proportion of neighboring data points that come from the flow direction; yellow regions show higher-than-average concentration of flow-direction $C_2$, blue regions show high concentration of vorticity-direction $C_2$, and dark/black regions show regions where $C_2$ are isotropically distributed. Since our selected heterodyne model states that the tell-tale fringe $C_2$ is linked to flow/

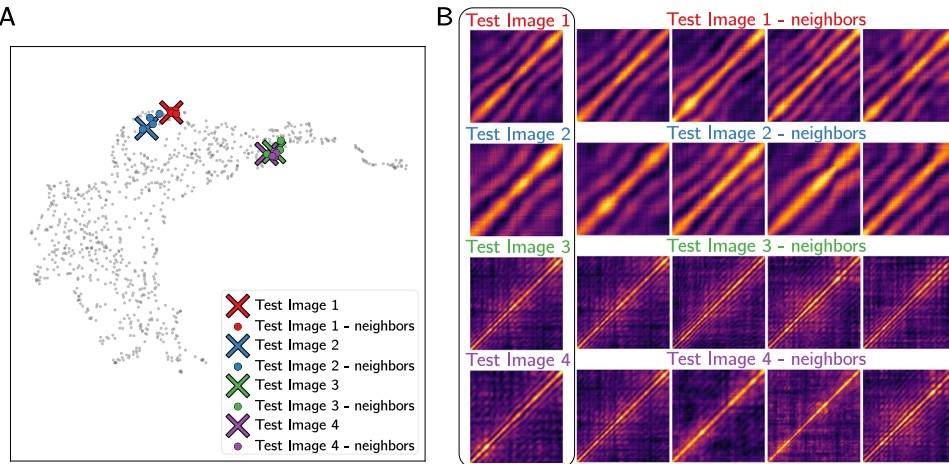

**Fig. 4 | Distances in the latent space are used to suggest similar images to user-specific images of interest.** In (**A**) sample heterodyne two-time correlations, $C_2$, are plotted in the Uniform Manifold Approximation and Projection (UMAP) visualization of the latent space as large X's. Nearest neighbors (calculated by Euclidean distance in the latent space) are shown as corresponding solid points. The UMAP visualization is cropped so that test data is easier to see. In (**B**) sample images are displayed alongside their nearest neighbors to evaluate similarity.

vorticity anisotropy, sampling from latent space regions which strongly correspond with specific scattering directions should allow us to confirm our hypothesis. Sampling from the blue region in the upper center of Supplemental Fig. 4 yields the target images in Fig. 4 which indeed show the characteristic heterodyne fringes. While to this point all analysis could be completed without having to view or interpret individual $C_2$, this type of ML-guided sampling allows us to verify our hypothesis that the data exhibits heterodyne scattering.

In addition to the difficulty of quantifying dynamics from XPCS data, another major bottleneck for the analysis of synchrotron scattering experiments is the amount of raw data that is collected, and then must be processed, reduced and analyzed. For context, advanced x-ray detectors used at APS can routinely collect 10–50 GB of raw scattering data per second. Many experiments may run continuously for hours producing terabytes of data from a single experiment, making manual and offline analysis impracticable. With this in mind, we demonstrate how our unsupervised latent space analysis can be used to easily explore immense experimental datasets. Since the autoencoder learns to recognize over-arching features of the entire data distribution, we can use the latent space distance between a user-specified target image and other points as a metric of similarity to identify other experimental conditions that produce the same behavior. In this case, our goal is to identify fringed heterodyne $C_2$, based on the model suggested above. Target images were selected to have similar overall appearance (all show fringed $C_2$), yet still represent significantly different behaviors: Images 1 and 2 show lower frequency fringes than Images 3 and 4, but the intensity along the diagonal band is unique in each test image. If our model accurately encodes $C_2$, all four test images should appear close together in the latent space, with Images 1-2 and Images 3-4 being even closer together. Fig. 4A shows each test image (marked with large X's) in the UMAP embedding of the latent representation, and the corresponding four nearest neighbors measured by euclidean distance in the latent space (the UMAP distribution is only used for visualization, all distances are calculated based on the AE encoding).

A major challenge with this approach of identifying patterns in the dataset by comparing distances between points is that many $C_2$ topologies only appear once, or a few times, even in very large datasets. While, our autoencoder can accurately capture and reproduce even the most unique and complicated patterns (see Supplemental Fig. 1 for a demonstration of $C_2$ reconstructions), it is still difficult to identify neighbors in the latent space for these complex data - the closest samples in latent space may still be very different from the

target of interest. This is demonstrated in Supplemental Fig. 5, where we see that the nearest neighbors for complex or rare patterns are less similar than the samples shown in Fig. 4.

Even though machine learning offers an opportunity to overcome bottlenecks in experimental data analysis, reliability, generalization to new data, and the danger of applying complex algorithms as a black box are areas of major concern for the application of machine learning in scientific data. In addition, the inner workings and results of the model can be difficult to interpret. With this in mind, ML results should be compared against analyses possible through simpler methods to evaluate the trade-off between increased accuracy at the cost of interpretability. To address this issue, we have performed additional tests to understand the benefits brought by studying our experimental system in the AI-NERD latent space, compared to performing similar analyses on the raw data. For the following analyses, shown in Supplemental Figs. 6, 7), AI-NERD refers to the 64-parameter latent space vector describing $C_2$, while the raw data representation refers to simply flattening the 256 x 256-pixel images to vectors of size $256^2$ such that both representations can be processed in similar ways. We find that clustering results and the appearance of the UMAP visualization are very similar regardless of $C_2$ representation, and therefore do not benefit from latent space analysis (Supplemental Fig. 6). This can be attributed to the fact that the main feature of the data is the mean intensity of $C_2$ corresponding to the average dynamics, and that clustering algorithms groups data by this dominant feature regardless of the dimensionality of the data set. That being said, AI-NERD shines in identifying similarities and differences between data, which is seen as the euclidean distance between $C_2$, rather than the assigned cluster label. To test the accuracy of AI-NERD similarity analysis versus analysis on raw image data, we manually identified a visually striking target image (Supplemental Fig. 7) and found the 300 most similar $C_2$ from a large dataset (13,000 unique $C_2$) using euclidean distance in raw image space, and the latent space. Three authors of this paper, who are experts in XPCS data collection and analysis but were not involved in ML model development or training, were asked to evaluate where the suggested $C_2$ was in fact similar to the target in a blind review process: each expert was handed two sets of 300 $C_2$, without knowing how they were obtained or labeled (see specific description in Section IV F). Experts were provided deliberately vague instructions, asking only whether $C_2$ is similar to the target without defining what features of the target were important; in this way, humans were essentially provided with the same information as the AI-NERD model. As shown by the

black curves in Supplemental Fig. 7 which represent the average of all expert evaluations, AI-NERD suggestions achieved greater than 50% accuracy for 260 out of the 300 suggested images, while raw data analysis only saw comparable accuracy on the 70 nearest neighbors (shown as vertical black lines in Supplemental Fig. 7). The differences in expert evaluations highlights the need for automated methods - this task is difficult and subjective even for trained experts, so automating data processing removes bias and variability. The drastic difference in performance between AI-NERD and raw data suggestions can be attributed to data sparsity, or the curse of dimensionality which states that distances between data points become increasingly indistinguishable as the dimensionality of the data increases. In summary, the ability of the latent space to represent complex relationships between data enables easier and accelerated exploration of data sets which will be crucial for experiments at next-generation light sources where the data production rates will make manual analysis impossible.

## Discussion

Our work has focused on encoding and categorizing full $C_2$, however, out-of-equilibrium systems exhibit multiple timescales, both within a single $C_2$, representing the milliseconds to seconds timescale, and across many $C_2$ in an entire experiment. Moreover, visual inspection of experimental non-equilibrium $C_2$ (for example, Fig. 1C) shows that dynamics in our glassy colloidal system are intermittent in many cases-rather than constantly changing dynamic characteristics, transitions appear as unpredictable changes between otherwise near-equilibrium states. Aside from temporal heterogeneity, spatial heterogeneity further complicates understanding non-equilibrium processes since the comparison of bulk scale measurements with XPCS measurements on a small area of the sample requires mechanistic descriptions that can cross length scales. In light of these challenges, understanding such transitions requires a systematic and unbiased method for observing and recognizing anomalous changes in large sets of data. Machine learning methods are an ideal choice for capturing subtle transitions while removing human bias, however, to date, applications of representation learning to scientific data focus on experimental techniques where data have discernible features, e.g., peaks with finite positions and widths. In contrast, our method focuses on using AI to identify important features in data that are difficult to interpret by eye, even for human experts. AI-NERD represents a step towards automatic recognition of fluctuations in time-resolved X-ray scattering, and understanding how these fluctuations relate to measurable properties. Future work could extend our method to not only cluster and track dynamics between $C_2$, but within sections cropped out of individual $C_2$ to investigate how relaxation behavior changes as a function of time-scale. Previous studies have demonstrated the fractal nature of relaxation processes in colloidal gels, meaning that structure and dynamics vary in a self-similar fashion across length and time scales. Our ability to track these changes, both within and across $C_2$, with statistical certainty will allow a greater understanding of how and why these mechanisms occur in disordered materials[74–76]. Interestingly, researchers have also identified fractal relationships between structure and dynamics in metallic glasses[77]. As our approach is not specific to studying colloidal suspensions and can be adapted for other classes of materials simply by retaining the CNN, this is another interesting case where our AI-NERD approach could be applied to understand relaxation across length and time scales.

The increasing rate of data production is an ubiquitous problem in nearly every field of technology. This is particularly a challenging problem for materials science and physics research, where scientific data often is high-resolution, time-resolved, and multi-dimensional making even data storage, without considering processing and analysis, challenging. Applications of machine learning to address these challenges have been wide-ranging in fields such as electron microscopy, high-energy physics, and synchrotron x-ray scattering. Despite

this progress, many scientific ML applications are limited to use in very specific types of experiments, data, or materials. ML also demands a significant barrier to entry, making adoption and adaptation of these techniques very difficult for scientists specializing in experimental research. Therefore, general scientific ML methods are required to, first, enable data analysis and discovery of new science in the age of big data, and, second, reach a wide audience to increase scientific productivity and creativity across fields. Our goal with AI-NERD is to develop a general workflow that uses AI to enhance real-time data analysis. Attention was paid to avoiding training the model in a way that would strictly apply to rheo-XPCS experiments (such as incorporating timescales in to the training data, or training on data occuring only under specific rheological conditions). As our model can accurately reproduce a wide range of $C_2$ topologies, we expect that application to other material classes should be as simple as fine-tuning the model on new data. Since training set generation involves simply placing all available experimental data into a single numpy array, with no need for curation and minimal preprocessing, the barrier to entry is reduced and scientists can focus on the results rather than issues with the training process. Moreover, in cases where less data is available, transfer learning from the pre-trained model should allow reasonable performance on new data. While, at this point, more work is required to extract quantitative physical information from the AI framework, our method for flexible and intuitive dataset exploration accelerates scientific research through the identification of areas of interest so that scientists can focus on understanding the system rather than wrangling data.

In summary, we presented an unsupervised procedure for the automation of XPCS data exploration and analysis. The workflow allows us to explore the structure and distribution of large experimental datasets that would be difficult to otherwise interpret. Moreover, unique visualizations allow us to understand the dynamics of an evolving system, build links between microstructural evolution and macroscopic properties in a way that is impossible using traditional data analysis, and select physical models to describe non-equilibrium dynamics. As characterization instrumentation continues to improve, the amount of data collected in a single experiment will grow exponentially, yet the amount of data that can be manually analyzed remains stagnant. Therefore, automation of as much of the data analysis process as possible is imperative to fully utilize modern experimental equipment. Our work using AI to guide the initial stages of data exploration and qualitative analysis represents an important step towards increasing the amount of available data that can be used and presents a framework for parsing large datasets. As each $C_2$ dataset is associated with many metadata parameters (such as collection time, position in the sample, viscosity, shear stress, volumetric concentration, particle size, etc.), visualization of the latent space is key for the explanation of the relationships between parameters. More importantly, this visualization and encoding framework is flexible and can be applied to experiments on other classes of materials, or even on different types of experimental data; while our analysis clearly shows how unsupervised deep learning can be used to link structure-property relationships across length scales in rheo-XPCS, our method is a generic image processing framework which requires no physical information and can therefore be applied to any experimental data which can be represented in two-dimensional/image space.

## Methods

### Rheology experiments
A sample of silica nanoparticles (200-300 nm) dispersed in polyethylene glycol (M.W. = 200) at a volume fraction of 60.5% is used to study the dynamics of glassy systems. The sample was loaded into a poly carbonate cylindrical Couette cell with a bob and cup (5.5 mm and 5.7 mm radii, respectively). The shear cell was driven by an Anton Paar MCR 301 rheometer. The X-ray beam is aligned at the center of the

shear cell, so the detector plane is in the $q_v$ - $q_{\Delta xv}$ direction. The sample is sheared under various conditions including preshear, steady shear ramp, and start-up shear. After the shear sequence, the shear rate was set to zero, and the XPCS experiments were conducted to monitor the dynamics of particles at various positions of the sample. Through XPCS measurements, the rheometer constantly monitors the stress relaxation process.

## X-ray Photon Correlation Spectroscopy on Silica Nanoparticle Glass

The XPCS measurement was performed at Beamline 8-ID-I of Advanced Photon Source, Argonne National Laboratory. An X-ray beam was generated by tandem 33 mm period, 2.4 m length undulators and was first deflected from a plane silicon mirror at an angle of incidence of 2.5 mrad and then filtered through a Ge(111) double-crystal monochromator with a relative bandpass of 0.03% to select a longitudinally coherent X-ray beam with a photon energy of 11 keV. The beam was then apertured horizontally to match the transverse coherence length at the entrance of the X-ray focusing optics (Beryllium Compound Refractive Lenses) and focused along the vertical direction, resulting in a 15 μm × 10 μm footprint on the sample with a total flux of $1.2 \times 10^{10}$ photons per second.

The scattered X-ray intensities were collected at a distance of 8 m from the sample using a Lambda 750 k photon-counting detector with 55 μm pixel size and 512 × 1536 pixels[78]. The XPCS analysis focuses on the region of detector pixels (Region of Interest, ROI) within the vicinity of the first peak in the structure factor (0.019 nm$^{-1}$ < $Q$ < 0.029 nm$^{-1}$), and the ROI was further partitioned into 18 smaller ROIs in the angular direction (20° width) to account for the azimuthal asymmetry of the dynamics resulting from the rheological shear. $C_2$ is calculated from the multiplication of normalized intensity fluctuation $D(\mathbf{Q}, t)$ averaged over the entire ROI[13,79]:

$$C_2(t_1, t_2) = \langle D(\mathbf{Q}, t_1) \cdot D(\mathbf{Q}, t_2) \rangle_{i,j} \qquad (1)$$

where $\langle \ldots \rangle_{i,j}$ indicate the pixel average. $D(\mathbf{Q}, t)$ is defined as:

$$D(\mathbf{Q}, t) = \frac{I(\mathbf{Q}, t) - \langle I(Q) \rangle_t}{\langle I(Q) \rangle_t} \qquad (2)$$

where $\langle I(Q) \rangle_t$ is the 1D Small-angle x-ray scattering (SAXS) intensity at the pixel with momentum transfer $\mathbf{Q}$, i.e., azimuthal average of the time-average from the detector frame sequence.

## Machine learning dataset construction

All $C_2$ in the dataset were measured on scattering patterns from silica sphere suspension at differing volume fractions and rheological conditions. All $C_2$ collected during one experimental time were aggregated to build an initial training set of 13,248 unique $C_2$. Experimental data were collected at 100 frames per second for 50 seconds, so raw $C_2$ are 5000 x 5000 pixels. Data were downsized to 256 x 256 for model training using the scikit-image resize function with gaussian anti-aliasing ($\sigma = 2$)[80]. After downsizing to 256 x 256 pixels, each $C_2$ was subject to data augmentation by 25 random shifts along the diagonal to increase the size of the dataset to 331,200 unique samples. Finally, before training all data were scaled so that intensities fell in the range [0,1]

## CNN autoencoder model

We used a standard hourglass-style convolutional neural network as our autoencoder architecture. This model uses three stages in both the encoding and decoding networks, where each stage consists of two convolutional layers, followed by dropout regularization (factor of 0.25), and ReLU activation. The number of features in each encoding stage was 8, 16, 16, and 32 (the decoding model uses the reverse). After

three convolutional stages, the data was flattened into a vector and passed through a fully connected layer. This produces the latent representation of data. A second fully connected layer is used to start the decoding process, and the output of this layer is transformed back into tensor form before passing to the decoder CNN. An exact description of CNN architecture is shown in Supplemental Fig. 8. To rapidly reduce the dimensionality of the data, and reduce the number of trainable parameters, we applied max-pooling to reduce the size of images by a factor of four after each stage; in the decoding model, it was found that upsampling after the convolutional layers performed better than using transpose convolution layers to upsample the images[81,82]. Increasing either the number of convolutional layers or the number of filters per layer was found to degrade the quality of output image reconstructions; even with the augmented dataset, model convergence was not stable as the size of the model increased. Similarly, including KL-divergence loss for latent space regularization (as used in training VAE) severely reduced the quality of reconstruction.

We trained models with latent dimensions varying from 2 - 1024 (increasing in powers of two) to optimize the expressive power of the latent representation. After training each model, the mean squared error was evaluated on a test data set and the mean of the error was plotted as a function of latent dimension. As shown in Supplemental Fig. 9, the error rapidly decreased and leveled off at a latent dimension of 16. We chose to use a bottleneck layer of size 64 for the final model to balance high accuracy with the complexity of the latent representation.

The model was trained on the 100-times augmented dataset for 60 epochs using a cyclic learning rate in the Pytorch DL framework[83]. Learning rate scheduling parameters are as follows: learning rate step size was defined to be $6N$, where $N$ is the number of mini-batches per training epoch, and the minimum and maximum learning rates were $1 \times 10^{-4}$ and $1 \times 10^{-3}$, respectively. The learning rate cycle was defined by the triangular2 mode in Pytorch. Mean squared error loss was used to optimize the weights. Scripts for defining and training these ML models can be found on GitHub.

## Clustering and visualization

After training the autoencoder, $C_2$ images were passed through the encoder stage only to produce the latent representation of the dataset. KMeans clustering was initially applied using the scikit-learn library with the number of clusters ranging from 2 to 12[84]. Plotting distortion and the silhouette score as a function of the number of clusters, the ideal number of clusters was determined to be in the range of four - six using the elbow method (Supplemental Fig. 3). In this work we have chosen to use the K-means clustering algorithm because of it's scalability to large data sets which enables rapid analysis and iteration. However, the choice of clustering algorithm must be made with regards to its underlying assumptions, and we note that when using AI-NERD in other applications which show strong structuring, non-uniformity, or large numbers of outlying data points in the latent space, the K-means algorithm may not be most effective. We performed systematic tests of other clustering algorithms, especially those which do not require the user-specified number of clusters, and found in general that these algorithms converge to an unrealistic number of clusters and depend heavily on hyperparameter tuning. Aside from the choice of the clustering algorithm, the choice of the number of clusters has the potential to impact the ensuing analysis. While the elbow method is commonly applied to determine the ideal number of clusters, the exact placement of the elbow can be subjective. We have found that our clusters correspond to the relative rate of dynamics in a nearly continuous distribution, and therefore variations in the number of clusters do not significantly impact our findings. We caution that in other applications where stronger clustering behavior is present, careful attention must be paid to clustering results and visualization. Plots

showing our systematic tests of clustering algorithms and cluster number choices can be found in our analysis code.

Samples from each cluster were drawn to evaluate similarity within each cluster. Uniform Manifold Approximation and Projection (UMAP) was used to project the 64-dimensional latent space into a two-dimensional visualization to inspect the quality of the clustering results and the position of optimized cluster centers. UMAP parameters were optimized manually, and we found that setting the number of neighbors and minimum distance parameters to 5, and 0.25, respectively, produced a visualization with clear trends, and which maintains similarity between $C_2$ in the same general region.

### Blind expert evaluation
The blind review was carried out as follows:

1. A target C2 pattern was chosen with specific fringe features of interest.
2. The 300 nearest neighbors to the target image were identified using both AI-NERD and Euclidean distance in the raw data space.
3. Data for each case were provided as an unlabeled PowerPoint slide deck to domain experts (Q.Z., S.N., E.D.), who then went through all 600 C2 suggestions and labeled them as 'similar' or 'not similar'. We emphasize the researcher who collected and prepared the data did not participate in the evaluation of the suggestions, and that the sets of C2 images were unlabeled such that it was impossible to tell whether suggestions came from AI-NERD or conventional analysis.

### Reporting summary
Further information on research design is available in the Nature Portfolio Reporting Summary linked to this article.

## Data availability
The data generated in this study and used for analysis have been deposited in the Zenodo database under https://doi.org/10.5281/zenodo.100059000[85]. Source data for plots presented in this work are provided with this paper. Source data are provided with this paper.

## Code availability
Python scripts for reproducing analyses presented in this paper are available in a GitHub repository with persistent https://doi.org/10.5281/zenodo.10022423[86,87].

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

## Acknowledgements

This research used resources of the Advanced Photon Source and Center for Nanoscale Materials, U.S. Department of Energy (DOE) Office of Science user facilities, and is based on work supported by Laboratory Directed Research and Development (LDRD) funding from Argonne National Laboratory, provided by the Director, Office of Science, of the U.S. DOE under Contract No. DE-AC02-06CH11357. W.C. and H.H. were supported on XPCS data collection and analysis by the U.S. Department of Energy, Office of Science, Office of Basic Energy Sciences, Materials Science and Engineering Division. S.K.R.S.S. and M.J.C. were partially supported by the US Department of Energy, Office of Science, Office of Basic Energy Sciences Data, Artificial Intelligence, and Machine Learning at DOE Scientific User Facilities program under Award Number 34532 (Digital Twins). We thank Nina Andrejevic and Saugat Kandel for helpful discussions on latent space analysis and CNN architecture optimization, respectively.

## Author contributions

All authors contributed to the conception of the research topic. Neural network development and training were performed by J.P.H. and M.J.C. and clustering analysis was performed by J.P.H. with input from M.J.C., Q.Z., and S.N. X-M.L., H.H., E.M.D, and S.N. collected experimental data, with assistance from M.C. for initial calculation of $C_2$. Q.Z., E.M.D., and S.N. served as human experts to evaluate AI-NERD accuracy. S.K.R.S.S. and W.C. provided domain expertize for relating AI-NERD results to physical insight. All authors contributed to the interpretation of the results and preparation of the manuscript.

## Competing interests

The authors declare that they have no competing interests.
