## [Peer Review File · Nature Communications]

AI-NERD: Elucidation of Relaxation Dynamics Beyond Equilibrium Through AI-informed X-ray Photon Correlation SpectroscopyREVIEWER COMMENTS

Reviewer #1 (Remarks to the Author):

Summary:

The authors report on an unsupervised ML workflow they've termed AI-NERD for the automated classification of experimental rheo-XPCS data. Their workflow utilizes a convolutional neural network (CNN) autoencoder to encode XPCS C2 image data to a 64-dimension latent space followed by the application k-means clustering to evaluate their dimensionally reduced encoded data. The encoded latent space is used as the basis for a similarity metric to identify experimental conditions in large datasets that produce the same behavior. I'm not familiar with the XPCS field/technique, but the authors appear to make important strides towards improved XPCS data interpretation and large XPCS dataset analysis by their use of deep learning. However, I have some questions about the data analysis.

Comments:

Page 4, first line, I think there is a typo: "later" should be "latent".

Unfortunately, I received a 404 error when attempting to access the Github page for CNN layer architecture and hyperparameters. The code repository on Github would be a nice resource, but easy to describe CNN details like layer descriptions, activations, hyperparameters, and loss functions should be included in the manuscript or SI.

The authors are using the CNN autoencoder to learn a latent representation of the C2 image data. The 64-dimension coordinates of this latent space serve as the input to their k-means clustering analysis and as a metric of similarity. The autoencoder design will be critical in determining the structure and properties of the latent space. Therefore, more discussion of different autoencoder architectures and the impact of different architectures on the properties of the learned latent space could be warranted.

CNN autoencoders are often used to denoise images and appear to have been used for this purpose on XPCS data in the literature. But this work focuses on the 64D learned latent representations for subsequent k-means clustering and a Euclidean similarity metric, not on the denoised images. So, this is primarily a latent representation learning focused use of an autoencoder. I wonder if the CNN autoencoder is an ideal choice for this representation learning task. The learned latent space is unlikely to be continuous, can the authors comment on this? Do points close together in the 64D latent space necessarily yield similar C2 images when decoded? Can this be systematically investigated using the decoder? If nearby points in the 64D latent space do not always generate similar C2 images, what impact might this have on their clustering and Euclidean distance similarity metric analysis?

With respect to examining the latent space, recent publications that apply different autoencoder structures to spectroscopic data (e.g., Routh et. al. J. Phys. Chem. Lett. 2021, 12, 2086–2094.; Grossutti et. al. J. Phys. Chem. Lett. 2022, 13, 5787–5793) and image data (e.g., Higgins et. al. 2017. beta-vae: Learning basic visual concepts with a constrained variational framework) for latent representation learning could be considered. In these examples, either principal component analysis is used to assess/transform the latent space or a variational autoencoder (VAE) architecture is used to ensure a continuous latent space. VAE-based architectures are well established for generative modeling and image data reconstruction and would yield a continuous latent space. Did the authors consider a convolutional VAE for their analysis? Do the authors see any benefit to learning a continuous latent space?

k-Means clustering analysis revealed 4-6 optimal clusters from the elbow method. Was the quality of the five clusters in the manuscript evaluated by any other method, such as silhouette score?

Different clustering algorithms are more/less suited to different internal data structures. Do other general classes of clustering algorithms, such as hierarchical or density-based clustering

algorithms yield similar results or different insights?

Is the clustering sensitive to the latent space dimensionality? As mentioned above I had some questions about the latent space properties of the CNN autoencoder. 64D seems like a completely reasonable choice, but I wonder if your analysis would change for 32D or 128D?

Reviewer #2 (Remarks to the Author):

The authors report on an unsupervised deep learning framework for classification of two-time correlation maps measured by XPCS and want to relate bulk properties to microscopic dynamics.

The manuscript addresses the bottle neck of large amount of collected data common to many SR techniques which will also increase with future upgrades of SR sources and X-ray detectors. Automated classification may help for a fast/real time feedback during experiments but also in the off-line analysis. The authors demonstrate quite convincingly that correlation maps can be classified using existing ML frameworks. As such the topic is timely and interesting and the results will be of interest to the XPCS community.

However, the manuscript does not provide enough new insights into sample dynamics and/or ML assisted X-ray physics to warrant publication in Nature Communications. The paper is merely a demonstration of the feasibility of classification of TTC maps which is not too surprising or new, given the known image classification power of ML methods. There is still a long way from here to a real physical understanding and this paper clearly does not accomplish the really important next step. Nevertheless, the methods and results are sound, and I recommend seeking publication in a more specialized journal.

A few more comments in detail which may help with resubmission

Page 2

"analysis of XPCS data can be difficult" – quite general remark

"...typical XPCS data assumes that dynamics does not change" – this is not state of the art. TTCs are used since more than a decade now

Page 3

The TTCs in the supplement appear to be much cleaner than the ones shown in the manuscript. It remains unclear how this affects the classification.

Page 8

"... maintain physical meaning.."

Why are the authors sure at this point that the physical interpretation remains the same within the groups, when before the physical interpretation of TTCs still seemed quasi impossible?

Fig 2: It would be interesting to see these outlier TTCs and how they behave compared to the rest of Class 4. Otherwise you could save the second dimension in the latent space because the data are already sorted along the y-axis to some extent.

Page 10

"relevant temporal features from time-resolved data"

It appears that the main information is whether the TTC decays fast or slow. It remains unclear how more complex features or a strongly heterogeneous dynamics would show up in the classes.

Page 10

"without requiring assumptions of physical models"

I guess this is the weakest point of the paper that it remains totally unclear how to connect the classes to physics.

Reviewer #3 (Remarks to the Author):

This paper tackles an important task, that of examining the dynamics of materials, specifically considering an application of a colloidal glass under shear, and shows how XPCS data can be correlated with rheometer data.

However, the novelty of the work is unclear to me. It is well understood how to use encoders-decoders to generate lower-dimensional latent-space representations of high-dimensional data. Since my own expertise is not in rheology of materials, I cannot comment on whether the application is novel in that field. If so perhaps a more specialised publication for that field of research would be more appropriate for this work?

Quite apart from this question of novelty, the work has two substantial limitations that I feel would need to be addressed before it could be considered for publication:

(1) The work is not reproducible, since important details are not included in the paper. For example, the details of the encoder-decoder architecture is not presented, there is no information on the training algorithm or hyperparameters. (A reference to Python scripts on a github site, which could be deleted or changed at any time, is insufficient.) On the other hand, the paper lists the specific make and model of the GPU, which is not a problem to include, but is of little or no relevance to reproducibility. This is a good article on the importance of transparency and reproducibility: <https://www.nature.com/articles/s41586-020-2766-y>

(2) The paper has some nice visualisations of embeddings, along with qualitative observations, but no empirical evaluation is presented to demonstrate that the technique proposed by the authors is better than alternatives from the literature. There are not even any evaluations against trivial "straw man" approaches. For example, how does the approach demonstrated in Fig 4 compare against using k-nearest neighbours on the original data as opposed to on the latent-space mapping of the data? Also, when presenting qualitative results such as those in Fig 4, it is good practice to show some examples of failures as well as successes, to avoid the appearance of cherry-picking results.

In addition, the paper has some minor weaknesses:

(1) The paper tends to use the terms "class/classification" and "cluster" interchangeably, though they are generally considered to be distinct concepts in the ML literature. For example, the clusters found by k-means should be called "Cluster 0" etc, not "Class 0".

(2) Figure 1(D) is described as "denoised", but there is no information presented to confirm that there is indeed noise in the cross-correlation visualisations.

(3) Figure 2 is described as showing clusters, but Classes 3 and 4 overlap, while Class 4 contains 3 completely separate groups.

(4) In Fig 3, it would be helpful to arrange the two figures with A on top of B, not to the left of B, so that the reader could see more easily how they align versus time.

(5) The discussion of Fig 3 is made difficult because of references to "green" and "dark green" bands, when in fact there are 3 shades of green and a never-mentioned blue colour.

(6) The discussion at the end of Section II is quite speculative in nature, with little evidence to support it.

(7) Reference 63 is incomplete.

A final observation: given that the data is continuous in UMAP space, and is not really arranged

into discrete clusters, I think it would be more interesting to fit a regression function to the latent space data, rather than clustering it.

REVIEWER COMMENTS

Reviewer #1 (Remarks to the Author):

Summary:

The authors report on an unsupervised ML workflow they've termed AI-NERD for the automated classification of experimental rheo-XPCS data. Their workflow utilizes a convolutional neural network (CNN) autoencoder to encode XPCS C2 image data to a 64-dimension latent space followed by the application k-means clustering to evaluate their dimensionally reduced encoded data. The encoded latent space is used as the basis for a similarity metric to identify experimental conditions in large datasets that produce the same behavior. I'm not familiar with the XPCS field/technique, but the authors appear to make important strides towards improved XPCS data interpretation and large XPCS dataset analysis by their use of deep learning. However, I have some questions about the data analysis.

Author response: We thank the referee for their detailed and comprehensive comments and questions. We appreciate their time spent and believe the manuscript is significantly improved as a result.

=====
Referee comment:

Page 4, first line, I think there is a typo: "later" should be "latent".

Author response: We thank the referee for pointing this out, we have made broad changes to the text including that portion.

=====
Referee comment:

Unfortunately, I received a 404 error when attempting to access the Github page for CNN layer architecture and hyperparameters. The code repository on Github would be a nice resource, but easy to describe CNN details like layer descriptions, activations, hyperparameters, and loss functions should be included in the manuscript or SI.

Author response: We apologize for the inconvenience. We mistakenly had the github repo set as private. This has been fixed, and it is accessible now. We agree that the model should be described in more detail. We have expanded the discussion of the architecture in the methods section and added a layer-by-layer description of the model in the Supplemental Information (Supplemental Figure 8).

Specifically, both the encoding and decoding stages of our CNN consist of three convolutional blocks, each of which uses two successive sets of convolution-dropout-ReLU, followed by max-pooling. In the encoding stage, we use 16, 32, and 64 features for each block, and in the decoding stage the number of features in each block is reversed. Between the encoding and decoding stages, the data is flattened, and passed through a bottleneck layer which defines the latent dimension (64 latent features in this case).

Layer (type)	Output Shape	Param #
Conv2d-1	[-1, 16, 256, 256]	160
Dropout-2	[-1, 16, 256, 256]	0
ReLU-3	[-1, 16, 256, 256]	0
Conv2d-4	[-1, 16, 256, 256]	2,320
Dropout-5	[-1, 16, 256, 256]	0
ReLU-6	[-1, 16, 256, 256]	0
MaxPool2d-7	[-1, 16, 64, 64]	0
Conv2d-8	[-1, 32, 64, 64]	4,640
Dropout-9	[-1, 32, 64, 64]	0
ReLU-10	[-1, 32, 64, 64]	0
Conv2d-11	[-1, 32, 64, 64]	9,248
Dropout-12	[-1, 32, 64, 64]	0
ReLU-13	[-1, 32, 64, 64]	0
MaxPool2d-14	[-1, 32, 16, 16]	0
Conv2d-15	[-1, 64, 16, 16]	18,496
Dropout-16	[-1, 64, 16, 16]	0
ReLU-17	[-1, 64, 16, 16]	0
Conv2d-18	[-1, 64, 16, 16]	36,928
Dropout-19	[-1, 64, 16, 16]	0
ReLU-20	[-1, 64, 16, 16]	0
MaxPool2d-21	[-1, 64, 4, 4]	0
Flatten-22	[-1, 1024]	0
Linear-23	[-1, 64]	65,600
ReLU-24	[-1, 64]	0
Linear-25	[-1, 1024]	66,560
ReLU-26	[-1, 1024]	0
Unflatten-27	[-1, 64, 4, 4]	0
Conv2d-28	[-1, 64, 4, 4]	36,928
Dropout-29	[-1, 64, 4, 4]	0
ReLU-30	[-1, 64, 4, 4]	0
Conv2d-31	[-1, 64, 4, 4]	36,928
Dropout-32	[-1, 64, 4, 4]	0
ReLU-33	[-1, 64, 4, 4]	0
Upsample-34	[-1, 64, 16, 16]	0
Conv2d-35	[-1, 64, 16, 16]	36,928
Dropout-36	[-1, 64, 16, 16]	0
ReLU-37	[-1, 64, 16, 16]	0
Conv2d-38	[-1, 64, 16, 16]	36,928
Dropout-39	[-1, 64, 16, 16]	0
ReLU-40	[-1, 64, 16, 16]	0
Upsample-41	[-1, 64, 64, 64]	0
Conv2d-42	[-1, 32, 64, 64]	18,464
Dropout-43	[-1, 32, 64, 64]	0
ReLU-44	[-1, 32, 64, 64]	0
Conv2d-45	[-1, 32, 64, 64]	9,248
Dropout-46	[-1, 32, 64, 64]	0
ReLU-47	[-1, 32, 64, 64]	0
Upsample-48	[-1, 32, 256, 256]	0
Conv2d-49	[-1, 1, 256, 256]	289
Sigmoid-50	[-1, 1, 256, 256]	0

Total params: 379,665
 Trainable params: 379,665
 Non-trainable params: 0

Input size (MB): 0.25
 Forward/backward pass size (MB): 81.27
 Params size (MB): 1.45
 Estimated Total Size (MB): 82.97

Figure R1. Copy of Supplemental Figure 8, showing the exact CNN architecture.

Changes in manuscript: Added supplementary figure 8 with detailed model summary. A description of the network architecture was added to the methods section (Section VD, copied below). We have also saved a persistent version of our model to zenodo (DOI:10.5281/zenodo.8101899) in addition to the updated public github repo (https://github.com/jhorwath/XPCS_Clustering)

“We used a standard hourglass-style convolutional neural network as our autoencoder architecture. This model uses three stages in both the encoding and decoding networks, where each stage consists of two convolutional layers, followed by dropout regularization (factor of 0.25), and ReLU activation. The number of features in each encoding stage was 8, 16, 16, and 32 (the decoding model uses the reverse). After three convolutional stages, the data was flattened into a vector, and passed through a fully connected layer. This produces the latent representation of data. A second fully connected layer is used to start the decoding process, and the output of this layer is transformed back into tensor form before passing to the decoder CNN. An exact description of CNN architecture is shown in Supplemental Figure 9. In order to rapidly reduce the dimensionality of the data, and reduce the number of trainable parameters, we applied max-pooling to reduce the size of images by a factor of four after each stage; in the decoding model, it was found that upsampling after the convolutional layers performed better than using transpose convolution layers to upsample the images. Increasing either the number of convolutional layers or the number of filters per layer was found to degrade the quality of output image reconstructions; even with the augmented dataset model convergence was not stable as the size of the model increased. Similarly, including KL-divergence loss for latent space regularization (as used in training VAE) severely reduced the quality of reconstruction.

We trained models with latent dimensions varying from 2 - 1024 (increasing in powers of two) to optimize the expressive power of the latent representation. After training each model, mean squared error was evaluated on a test data set and the mean of the error was plotted as a function of latent dimension. As shown in Supplemental Figure 10, the error rapidly decreased and leveled off at a latent dimension of 16. We chose to use a bottleneck layer of size 64 for the final model to balance high accuracy with the complexity of the latent representation.

The model was trained on the 100-times augmented dataset for 60 epochs using a cyclic learning rate in the Pytorch DL framework. Learning rate scheduling parameters are as follows: learning rate step size was defined to be $6N$, where N is the number of mini-batches per training epoch, the minimum and maximum learning rates were 1×10^{-4} and 1×10^{-3} , respectively. Learning rate cycle was defined by the *triangular2* mode in Pytorch. Mean squared error loss was used to optimize the weights. Scripts for defining and training these ML models can be found on github.”

=====

Referee comment:

The authors are using the CNN autoencoder to learn a latent representation of the C2 image data. The 64-dimension coordinates of this latent space serve as the input to their k-means clustering analysis and as a metric of similarity. The autoencoder design will be critical in determining the structure and properties of the latent space. Therefore, more discussion of different autoencoder architectures and the impact of different architectures on the properties of the learned latent space could be warranted.

CNN autoencoders are often used to denoise images and appear to have been used for this purpose on XPSC data in the literature. But this work focuses on the 64D learned latent representations for

subsequent k-means clustering and a Euclidean similarity metric, not on the denoised images. So, this is primarily a latent representation learning focused use of an autoencoder. I wonder if the CNN autoencoder is an ideal choice for this representation learning task. The learned latent space is unlikely to be continuous, can the authors comment on this? Do points close together in the 64D latent space necessarily yield similar C2 images when decoded? Can this be systematically investigated using the decoder? If nearby points in the 64D latent space do not always generate similar C2 images, what impact might this have on their clustering and Euclidean distance similarity metric analysis?

Author response: We thank the referee for this insightful comment. It is true that one of the main issues with standard autoencoders for representation learning is that the latent space is discontinuous and that the latent features are not orthogonal. Other models, such as variations on variational autoencoders (VAE) aim to solve this, however we found that the additional training constraints needed to enforce continuity in the latent space severely degraded our training accuracy. Since we are not aiming to use our CNN as a generative model, where feature-independence in the latent space is critical, we chose to use a standard autoencoder architecture to maximize reconstruction accuracy.

Upon the reviewer's suggestion we have performed tests to show that samples in the latent space do, in fact, correspond to similar images. To produce Supplemental Figure 2 we have selected a target real C2, and generate artificial C2 by adding small amounts of gaussian noise to the real latent vector. We can see that this process produces artificial C2 which appear similar to the target.

Figure R2. Copy of Supplemental Figure 2, showing generative capabilities of our AE model.

In the above figure (Figure R2, copy of Supplemental Figure 2), the large C2 on the left is a randomly chosen experimental C2, while the grid on the right shows 16 artificial C2 produced by the addition of gaussian noise to the latent vector of the target C2. The main features of the target data are retained across the 16 artificial samples: high intensity at the lower left, followed by a narrow intensity band, and finally a continually broadening intensity band. Some features in the artificial samples show complex dynamics not present in the target images (“wing shapes” in the third row, right two C2), yet are found in the broad C2 training dataset. This suggests that the model is capable of representing a wide range of

features, and that the latent space retains physical information even without enforcing continuity or orthogonality.

Changes in manuscript: Added supplementary figures 2 (copied above as Figure R2) to demonstrate that points close in the latent space yield qualitatively similar images when decoded. Main text changes can be found in the last two paragraphs of Section IIA. Here we discuss how our model was used probe the structure of the latent space, and how we reached our decision to use a standard autoencoder model. This discussion is copied below:

“To ensure that the latent space encoding accurately represents the distribution of experimental data, we used the trained model to generate artificial C2 by adding noise to the latent representation of real C2. In the ideal case, passing these artificial latent vectors through the decoder model will produce new data which is similar to the original C2. Supplemental Figure 2 demonstrates that our trained model is able to generate realistic data at a variety of noise levels, suggesting that the latent space is representative of the data. Here, 16 unique artificial C2 were generated by adding Gaussian noise with $\sigma = 0.3$ to the target latent vector - $\sigma = 0.3$ represents the average standard deviation of all latent variables for the encoded test dataset.”

=====

Referee comment:

With respect to examining the latent space, recent publications that apply different autoencoder structures to spectroscopic data (e.g., Routh et. al. J. Phys. Chem. Lett. 2021, 12, 2086–2094.; Grossutti et. al. J. Phys. Chem. Lett. 2022, 13, 5787–5793) and image data (e.g., Higgins et. al. 2017. Beta-vae: Learning basic visual concepts with a constrained variational framework) for latent representation learning could be considered. In these examples, either principal component analysis is used to assess/transform the latent space or a variational autoencoder (VAE) architecture is used to ensure a continuous latent space. VAE-based architectures are well established for generative modeling and image data reconstruction and would yield a continuous latent space. Did the authors consider a convolutional VAE for their analysis? Do the authors see any benefit to learning a continuous latent space?

Author response: We thank the referee for pointing us to these papers. As mentioned above, we did consider using a VAE model but found that it was not able to accurately reproduce C2 data. We attributed this to the inability of the model to capture the more complicated and widely varying off-diagonal features while being subjected to additional training/loss constraints.

We do not believe that a continuous latent space would influence the conclusions drawn in our manuscript: we have demonstrated that distance in our discontinuous latent space is enough to identify similar real C2 and generate similar artificial C2 (Figure R2, and Figure 4 in the main text), and that the broad clusters in our latent space correspond with physical dynamics and can be tracked through time (Figure 2 and 3 of the main text).

That being said, a continuous latent space with interpretable features would likely be interesting for future work focusing on relationships and transformations between specific spatial features in C₂, and may enable understanding complicated C2 patterns in terms of simpler “basis data”. For this type of

generative model, latent space continuity is important, however we believe that this is not an issue for the present analysis since we don't use the AE as a generative model.

Changes in manuscript: Added Supplemental Figure 2 showing images generated by decoding new points from the latent space. We have also added discussion to the main text covering the use of autoencoders and representation learning in material science (including mention of the suggested papers):

“To ensure that the latent space encoding accurately represents the distribution of experimental data, we generated artificial data by adding noise to the latent representation of real C2. In the ideal case, passing these vectors through the decoder model will produce new data which is similar to the original C2. Supplemental Figure 2 and 3 demonstrates that our trained model is able to generate realistic data at a variety of noise levels, suggesting that the latent space is representative of the data. In Supplemental Figure 2, 16 unique artificial C2 were generated by adding Gaussian noise with $\sigma = 0.3$ to the target latent vector - $\sigma = 0.3$ represents the average standard deviation of all latent variables for the encoded test dataset.

A major concern in the field of representation learning is the development of ML models which enforce continuity and orthogonality in the latent space to produce latent dimensions which are independent and may directly relate to physical features. While these approaches have proven successful in the x-ray characterization papers mentioned above, we have found that using such regularization schemes did not allow us to produce accurate reconstruction of C2 during the training process. We have chosen to use a standard convolutional autoencoder, in contrast to a variational autoencoder (VAE) or other type of model, to maximize the amount of information our model can learn at the expense of a discontinuous latent space.”

Referee comment:

k-Means clustering analysis revealed 4-6 optimal clusters from the elbow method. Was the quality of the five clusters in the manuscript evaluated by any other method, such as silhouette score?

Author response: We thank the referee for this suggestion. We have tested the silhouette similarity metric and found that the silhouette score suggests 4-6 clusters as the optimal number since after 6 clusters the silhouette score rapidly decreases. Silhouette scores range from -1 to 1, with 1 representing ideal cluster separation, values near 0 representing overlapping clusters, and negative values showing that samples have been assigned to the wrong cluster. Therefore, the sudden drop in silhouette score near 4-6 clusters represents the largest number of separable clusters seen in the dataset. The silhouette score has been added to supplemental figure 3 (Figure R3, below).

Figure R3. Copy of Supplemental Figure 3, showing elbow plots for choosing the optimal number of k-means clusters using clustering inertia and silhouette score metrics.

Changes in manuscript: Updated supplementary figure 3 (copied above) to include the silhouette score, and mentioned the use of the silhouette metric in the methods section (Section V, E).

“After training the autoencoder, C2 images were passed through the encoder stage only to produce the latent representation of the dataset. KMeans clustering was initially applied using the scikit-learn library with the number of clusters ranging from 2 to 12. Plotting distortion and the silhouette score as a function of number of clusters, the ideal number of clusters was determined to be in the range of four - six using the elbow method (Supplemental Figure 3).”

=====

Referee comment:

Different clustering algorithms are more/less suited to different internal data structures. Do other general classes of clustering algorithms, such as hierarchical or density-based clustering algorithms yield similar results or different insights?

Author response: We have tried clustering with various algorithms. For density-based clustering, we tested DBSCAN and found that regardless of hyperparameter choice we could not achieve clustering which matched trends in our data. We found that spectral clustering methods produced results similar to those in K-Means, while agglomerative clustering was not able to separate the data. We used K-Means since it is well known and easy to understand.

Figure R4. Comparison of clustering results using different algorithms

Referee comment:

Is the clustering sensitive to the latent space dimensionality? As mentioned above I had some questions above the latent space properties of the CNN autoencoder. 64D seems like a completely reasonable choice, but I wonder if your analysis would change for 32D or 128D?

Author response: We thank the referee for this insightful comment.

We expect that the cluster results would not change as a function of the latent space dimension. Since the clusters have blurry boundaries, while the exact position of the cluster center/edge may change with latent dimension we don't expect different behavior for analysis like Figure 3 of the main text. We have found that the main feature used for clustering encoded C2 turns out to be the mean image intensity – as this is the most basic feature of the input data, the cluster results should remain the same even if the latent dimension is severely reduced or expanded. This is similar to Supplemental Figure 6, where we compare clustering on the 64-D latent space to cluster in 256^2 -D image space. That being said, we expect that the latent dimension is crucial for capturing more subtle features in the topology and appearance of C2, such as width of diagonal bands, appearance and position of dynamic positions, and off-diagonal features, which are crucial for our similarity analysis. Further evidence for this is seen in Supplemental Figure 10 (copied here as Figure R5), where we plot mean-squared reconstruction error on a test dataset as a function of latent space size. This plot shows an elbow at a latent space dimension of 16. We interpret this to say that 16 latent dimensions are critical to capture the wide range of C2 topologies, and further increasing the dimensionality allows the network to reproduce finer, rarer C2 features. Therefore, we maintain our view that 64D latent space balances simplicity required for clustering analysis with the ability to capture complex C₂ features for similarity analysis.

Figure R5. Copy of Supplemental Figure 9, showing the trend of mean squared error after training as a function of latent space dimension. The blue line shows the mean error on the test dataset, while the blue band represents the standard deviation.

=====

Referee comment:

Reviewer #2 (Remarks to the Author):

The authors report on an unsupervised deep learning framework for classification of two-time correlation maps measured by XPCS and want to relate bulk properties to microscopic dynamics.

The manuscript addresses the bottle neck of large amount of collected data common to many SR techniques which will also increase with future upgrades of SR sources and X-ray detectors. Automated

classification may help for a fast/real time feedback during experiments but also in the off-line analysis. The authors demonstrate quite convincingly that correlation maps can be classified using existing ML frameworks. As such the topic is timely and interesting and the results will be of interest to the XPCS community.

However, the manuscript does not provide enough new insights into sample dynamics and/or ML assisted X-ray physics to warrant publication in Nature Communications. The paper is merely a demonstration of the feasibility of classification of TTC maps which is not too surprising or new, given the known image classification power of ML methods. There is still a long way from here to a real physical understanding and this paper clearly does not accomplish the really important next step. Nevertheless, the methods and results are sound, and I recommend seeking publication in a more specialized journal.

Author response: We have made significant changes to the manuscript to address the feedback from all of the referees. In particular, we have

1. Reframed the paper to address pressing challenges in materials characterization. In addition to developing a physical understanding of complex systems, we highlight the need for automated and reliable data analysis as experimental instruments continue to improve and the burden of large datasets continues to grow.
2. Expanded scientific analysis of relaxation dynamics in complex fluids. We have heavily modified Figure 3 in the main text, and the surrounding discussion, to account for dynamic anisotropy in the system. We have also discussed how AI-informed visualizations can guide the selection of physical models to describe complex experimental data and discover new behavior, namely the presence of heterodyne scattering in our data which has specific microstructural implications mentioned in the text.
3. Added significant discussion of how our results directly impact other science cases across soft- and hard-matter physics, and expanded the discussion of future work to explain the applicability of our method.

We agree with the referee's assessment of the previous version of the manuscript, and we believe the changes we have made address the shortcomings pointed out by the referee.

=====

Referee comment:

A few more comments in detail which may help with resubmission

Page 2

"analysis of XPCS data can be difficult" – quite general remark

"...typical XPCS data assumes that dynamics does not change" – this is not state of the art. TTCs are used since more than a decade now

Author response: We agree with the referee and have removed this verbiage.

Changes in manuscript: Removed text as suggested by referee.

Page 3

The TTCs in the supplement appear to be much cleaner than the ones shown in the manuscript. It remains unclear how this affects the classification.

Author response: Supplemental Figure 1 shows output of the CNN (right) compared to the input (left). We expect that the CNN output appears cleaner, since random fluctuations at the pixel scale will not be learned in the training process. We have added black lines to this figure to highlight that we are showing pairs of raw/reconstructed images to help highlight that each pair of images represents raw input and corresponding CNN output. Moreover, the training dataset captures a wide range of experimental data collected under various experimental conditions, such that we expect pixel level noise to have a negligible impact on the reconstruction.

Changes in manuscript: Adjusted Supplemental Figure 1 to clarify that clean C2 are actually output reconstructions from the trained CNN.

Figure R6. Copy of Supplemental Figure 1, showing a comparison of randomly selected experimental C2 (left in each box), and corresponding CNN reconstructions (right).

=====

Referee comment:

Page 8

“... maintain physical meaning..”

Why are the authors sure at this point that the physical interpretation remains the same within the groups, when before the physical interpretation of TTCs still seemed quasi impossible?

Author response: We acknowledge the apparent contradiction with our previous statements, and that “physical meaning” can be ambiguous. Our statement about the difficulty interpreting TTCs was made specifically with regards to complex fluctuation and non-equilibrium shapes/features. We believe that basic, qualitative interpretation (“C2 A displays faster dynamics than C2 B”) is still possible – and this is what we aim to capture in our work. We have addressed this in the text to clarify our stance on what is/is not possible.

Changes in manuscript:

The line in question has been changed to: “Further, the nearly continuous transition between clusters illustrates that groupings defined by the k-means algorithm contain physically similar data, and that the clusters are capable of recognizing the physical trend of continuously changing relaxation rates.”

Figure 3 and the surrounding discussion in section II D have been extensively reworked to describe how physical information can be obtained from our latent space visualizations:

“Bringing us one step closer to our goal of bridging information across length scales, our first test aims to understand how fluctuations in rheological measurements correspond with the evolution of the structure and local dynamics of a complex fluid. Full experimental details can be found in Section Methods. Briefly, after forcing the system to flow under a high shear rate, the applied shear is removed and the material is allowed to relax to study the impact of non-equilibrium deformation on flow in the complex fluid; simultaneous rheological and x-ray scattering measurements are collected during the relaxation process to track the mechanical response and structural changes, respectively. The rheological data describing relaxation (Figure 3A) clearly shows non-monotonic behavior, however since the complex C2 patterns seen in experimental data preclude direct quantification of the dynamics, it remains impossible to link these macroscopic changes to microscopic dynamics. To address this, Figure 3B shows a time-resolved histogram of cluster labels describing the progression of average dynamics through time. The histogram is built by clustering C2 collected from each azimuthal bin (Figure 3B) at each time step, and tabulating the results as vertical, color-coded bars. Vertical black lines in both panels show the times of shear stress minima. The growing peaks of slow dynamics (low cluster numbers) seen between shear stress minima indicate that tracking material behavior through latent space encoding is able to tie intermittent microscopic dynamics to macroscopic rheological changes. We note that the dark peaks seen in Figure 3B correspond with increases in shear stress are also seen at later times ($t = 2000$ and 2500 s) where we see no changes in the bulk shear stress measurement. Since the incident x-rays probe local dynamics while rheology captures the average behavior of the macroscopic system, the presence of these peaks highlights that XPCS captures local dynamics which happen within the scattering volume of the incident x-ray beam but may not manifest in the macroscopic response. The ability to understand spatial heterogeneity and its impact on rheology would have significant impact on our understanding of jamming and viscoelasticity in colloidal suspensions, especially given our discovery that multiple dynamic components exist within regions the size of the x-

ray probe coherent volume. Many physical systems ranging from granular materials in industrial systems to geological motion and earthquakes depend on intermittent, avalanche-like dynamics which may not be present in measurements of average properties of the system. Thus, the impact of data-driven methods to dynamics which link anomalies with their physical response across length and time-scales is far reaching.

Analysis of cluster label as a function of azimuthal scattering angle provides more information about the system. Figure 3C displays a heatmap of cluster labels as a function of azimuthal bin (Figure 1B) and experimental time. Following the dashed red line provided to guide the eye, we observe that dynamics are anisotropic, highlighting the presence of directed motion. Through the cluster labels, we see that dynamics initially slow down in the flow direction (parallel to the applied shear, bins 5 and 14) with comparatively faster dynamics in the vorticity direction perpendicular to the applied shear (bins 1, 9, 10, and 18). Finally, in Figure 3D we observe the distribution of dynamics throughout the experiment by selecting three experimental time points, and projecting all 18 azimuthal C2 associated with each time point into the latent dimension UMAP plot. We see that, in agreement with Figure 3C, initial stages show tight groups of similar dynamics for all C2 collected at a given time, while as time progress the distribution in the latent space spreads drastically. Since our ML model learns to encode spatial features of C2 into the latent space, the position and spread of latent space points corresponds to the appearance of C2. The wide latent space distribution from points collected at later times suggests that, in addition to the angular dependence of relaxation rate, the visual appearance of C2 differs significantly with scattering angle. Results from analysis in Figures 3A-D suggest that: 1. Dynamics slow down through time, initiated in the flow direction, 2. the intermittent microscopic dynamics are tied to macroscopic rheological fluctuations (however, not all microstructural changes are reflected in the macroscopic response), and 3. in addition to the rate of dynamics, the appearance of C2 differs drastically depending on scattering angle.

Research studying dynamics in materials generally aims to define a characteristic time scale which describes the evolution mechanism. While XPCS is capable of precisely capturing material dynamics by measuring the decorrelation time between successive scattering frames, a physical model of the process is still required to extract quantitative information describing, for example, particle motion or relaxation rate in the sample. Model selection is difficult for non-equilibrium systems, where either physical models do not exist, or the selection of a model is complicated by the presence of rapid unexpected changes in dynamics. Furthermore, most theoretical frameworks describe the mean behavior of a system, yet the nature of XPCS experiments probes local dynamics which may differ from expected behavior on experimental time scales, selection of a model based on visual inspection of a few sample C2 from an entire experiment is difficult and potentially unreliable. Therefore, the ability to categorize different types of XPCS C2 patterns to guide the development of structural and dynamics models for evolving materials is crucial.

Considering the conclusions drawn from analysis of Figure 3 guides us towards the selection of the heterodyne scattering model to describe our data, which occurs when a difference in velocity between two or more dynamically distinct components in the system leads to constructive/destructive interference in correlation patterns, and enables the extraction of system-independent information about the relative velocity between components. Heterodyne scattering produces a uniquely recognizable fringe pattern in XPCS C2 in specific scattering directions. This phenomenon has been documented in some experimental systems, however it has not been observed in experiments of colloidal glasses. Thus, our AI-guided model selection provides unique insight into microstructure

evolution in complex materials, and enables future research to quantify the properties and dynamics in this system.

Supplemental Figure 4 shows the UMAP latent space, where each pixel is colored by the proportion of neighboring data points which come from the flow direction; dark blue regions show high concentration of flow-direction C2, dark red regions show high concentration of vorticity-direction C2, and light/white regions show regions where C2 are isotropically distributed. Since our selected heterodyne model states that the tell-tale fringe C2 are linked to flow/vorticity anisotropy, sampling from latent space regions which strongly correspond with specific scattering directions should allow us to confirm our hypothesis. Sampling from the dark red region in the upper center of Supplemental Figure 4 yields the target images in Figure 4 which indeed show the characteristic heterodyne fringes. While to this point all analysis could be completed without having to view or interpret individual C2 this type of ML-guided sampling allows us to verify our hypothesis that the data exhibits heterodyne scattering.”

Figure R7. Copy of Figure 3 in the manuscript. This figure has been significantly changed to show the dependence of cluster labels on azimuthal correlation bin.

Referee comment:

Fig 2: It would be interesting to see these outlier TTCs and how they behave compared to the rest of Class 4. Otherwise you could save the second dimension in the latent space because the data are already sorted along the y-axis to some extent.

Author response: We thank the referee for this intriguing comment. The UMAP projection (in 2D) is used solely as a visualization tool, and distances within the projection can only be evaluated qualitatively. The clustering is done on the 64D latent space. Hence, the outlying clusters in the UMAP visualization are still grouped in Cluster 4 even though they may appear far apart from the rest of Cluster 4 in the reduced UMAP space. This is now discussed in Section II, C of the manuscript.

Visually, the outlier C2 are nearly identical to the data seen, e.g. at 55s in Figure R7 in this document. They show nearly instantaneous decorrelation, and occur in the early stages of the experiment when motion due to the applied shear is still very fast.

While we note in the manuscript that UMAP has the advantage of stability with respect to random initialization over the tSNE embedding algorithm, the appearance of the embedding manifold is still based on a randomized optimization process. Therefore, while it may appear that the removal of outliers would enable discarding the second UMAP dimension and representing C2 with a single characteristic number, in reality this trend may be less apparent if we were to re-run the UMAP algorithm. We expect that UMAP results are reproducible in terms of trends and general relationships within the data, however any regression analysis may be misleading.

Referee comment:

Page 10

“relevant temporal features from time-resolved data”

It appears that the main information is whether the TTC decays fast or slow. It remains unclear how more complex features or a strongly heterogeneous dynamics would show up in the classes.

Author response: We thank the referee for highlighting this. After careful consideration of the referee’s statement, we recognize that we are actually looking at dynamical heterogeneity between C2, rather than within individual C2. We have added discussion in the text about how non-equilibrium systems display multiple characteristic time scales. We acknowledge that more work is required to understand fluctuations within C₂ at the ms-s timescale, but believe this method can still capture intermittent dynamic fluctuations between C₂ at the seconds-hours timescale. We have adjusted the text to reflect this.

Changes in manuscript: We have added to the discussion section (Section III) to discuss the presence of transient dynamics at multiple time scales, and how these likely correspond to different evolution mechanisms in the material. We also discuss future work which will aim at understanding fluctuations within C2.

“Our work has focused on encoding and categorizing full C2, however out-of-equilibrium systems exhibit multiple timescales, both within a single C2, representing the milliseconds to seconds timescale, and across many C2 in an entire experiment. Moreover, visual inspection of experimental non-equilibrium C2 show that dynamics in our glassy colloidal system are intermittent in many cases- rather than constantly changing dynamic characteristics, transitions appear as unpredictable changes between otherwise near-equilibrium states. Aside from temporal heterogeneity, spatial heterogeneity further complicates understanding non-equilibrium processes since comparison of bulk scale measurements with XPCS measurements on a small area of the sample requires mechanistic descriptions which can

cross length scales. In light of these challenges, understanding such transitions requires a systematic and unbiased method for observing and recognizing anomalous changes in large sets of data. Machine learning methods are an ideal choice for capturing subtle transitions while removing human bias, and our method represents a step towards automatic recognition of fluctuations in time-resolved x-ray scattering, and understanding how these fluctuations relate to measurable properties. Future work could extend our method to not only cluster and track dynamics between C2, but within sections cropped out of individual C2 to investigate how relaxation behavior changes as a function of timescale. Previous studies have demonstrated the fractal nature of relaxation processes in colloidal gels, meaning that structure and dynamics vary in a self-similar fashion across length and time scales. Our ability to track these changes, both within and across C2, with statistical certainty will allow greater understanding of how and why these mechanisms occur in disordered materials. Interestingly, researchers have also identified fractal relationships between structure and dynamics in metallic glasses. As our approach is not specific to studying colloidal suspensions and can be adapted for other classes of materials simply by retaining the CNN, this is another interesting case where our AI-NERD approach could be applied to understand relaxation across length and time scales.”

=====

Referee comment:

Page 10

“without requiring assumptions of physical models”

I guess this is the weakest point of the paper that it remains totally unclear how to connect the classes to physics.

Author response: As our classifications correspond to the rate of correlation decay, we can directly relate this to the physics events in the system. We can then correlate those events to physical stimuli imposed on the system. We have adapted Figure 3 in the manuscript (copied as Figure R7 in this document) and corresponding discussion to show how analysis of clustering as a function of both time and scattering wave vector can give immediate interpretation and guide the selection of a physical model to quantify dynamics. We have demonstrated how combining analysis of Figures 3A-D tells us that the system experiences intermittent dynamics, which slow significantly faster in the direction parallel with the driving, and which appear as visually distinct depending on the direction. Based on these analyses, we can immediately tie the observed dynamics to the heterodyne model of scattering from disparate suspension components with minimal human input. We note that this is the first, to our knowledge, identification of this type of dynamics in this system, highlighting the strength of our method to track and identify non-equilibrium dynamics.

We acknowledge that our method does not provide direct physical insight and have modified the text to reflect this. Our vision is that this method would be applied as an initial analysis step to understand, for example, what broad type of dynamics coincide with unexpected measurements to guide further, more in-depth quantification. Along these lines, our own future work aims at quantifying heterodyne dynamics, now that this phenomenon has been discovered in our system.

Changes in manuscript:

The discussion around figure 3 (Figure R7 above) has been extensively revised to give a more thorough and concrete physical picture of the system. Exact changes are copied above in the discussion of Figure R7.

=====

Referee comment:

Reviewer #3 (Remarks to the Author):

This paper tackles an important task, that of examining the dynamics of materials, specifically considering an application of a colloidal glass under shear, and shows how XPCS data can be correlated with rheometer data.

However, the novelty of the work is unclear to me. It is well understood how to use encoders-decoders to generate lower-dimensional latent-space representations of high-dimensional data. Since my own expertise is not in rheology of materials, I cannot comment on whether the application is novel in that field. If so perhaps a more specialised publication for that field of research would be more appropriate for this work?

Quite apart from this question of novelty, the work has two substantial limitations that I feel would need to be addressed before it could be considered for publication:

(1) The work is not reproducible, since important details are not included in the paper. For example, the details of the encoder-decoder architecture is not presented, there is no information on the training algorithm or hyperparameters. (A reference to Python scripts on a github site, which could be deleted or changed at any time, is insufficient.) On the other hand, the paper lists the specific make and model of the GPU, which is not a problem to include, but is of little or no relevance to reproducibility. This is a good article on the importance of transparency and reproducibility: <https://www.nature.com/articles/s41586-020-2766-y>

Author response:

We thank the reviewer for pointing this out – we agree that transparency and reproducibility are important, particularly when using ML techniques which could be applied as black box model in the future. We have added a verbal description of the architecture in section V, D, and have also copied the detailed architecture directly from PyTorch in Supplemental Figure 8 (see Figure R1, above). In addition, we have archived the Github repo through Zenodo and a included a persistent identifier (DOI: 10.5281/zenodo.8101899).

Changes in manuscript:

A full layer-by-layer description of the model has been added as Supplemental Figure 9 (copied as Figure R1, above), and text has been added to the Methods Section V, D.

“We used a standard hourglass-style convolutional neural network as our autoencoder architecture. This model uses four stages in both the encoding and decoding networks, where each stage consists of two convolutional layers, followed by dropout regularization (factor of 0.25), and ReLU activation. The number of features in each encoding stage was 8, 16, 16, and 32 (the decoding model uses the reverse). After four convolutional stages, the data was flattened into a vector, and passed through a fully

connected layer. This produces the latent representation of data. A second fully connected layer is used to start the decoding process, and the output of this layer is transformed back into tensor form before passing to the decoder CNN. An exact description of CNN architecture is shown in Supplemental Figure 9. In order to rapidly reduce the dimensionality of the data, and reduce the number of trainable parameters, we applied max-pooling to reduce the size of images by a factor of four after each stage; in the decoding model, it was found that upsampling after the convolutional layers performed better than using transpose convolution layers to upsample the images. Increasing either the number of convolutional layers or the number of filters per layer was found to degrade the quality of output image reconstructions; even with the augmented dataset model convergence was not stable as the size of the model increased. Similarly, including KL-divergence loss for latent space regularization (as used in training VAE) severely reduced the quality of reconstruction.”

=====

Referee comment:

(2) The paper has some nice visualisations of embeddings, along with qualitative observations, but no empirical evaluation is presented to demonstrate that the technique proposed by the authors is better than alternatives from the literature. There are not even any evaluations against trivial “straw man” approaches. For example, how does the approach demonstrated in Fig 4 compare against using k-nearest neighbours on the original data as opposed to on the latent-space mapping of the data? Also, when presenting qualitative results such as those in Fig 4, it is good practice to show some examples of failures as well as successes, to avoid the appearance of cherry-picking results.

Author response:

We agree with the referee and thank them for pointing out this lack of baselines in our previous draft. We have now included comparisons to clustering on the original data as well as examples of failures.

We have added a paragraph in the discussion section to describe additional tests we have devised to evaluate the efficacy of our method in comparison to performing similar analyses on the raw data in a more traditional fashion. These tests are associated with Supplemental Figures 6-8, which compare AI clustering results with similar analysis applied to the raw data (taking a flattened version of our input data as a $N \times (256^2)$ -dimensional dataset).

While UMAP embedding and clustering results is similar between ML encoded data and the raw data (Figure R6 below), the main advance enabled by our machine learning method is significantly greater accuracy in determining similarities between data points – only through latent space analysis can we capture similarities and differences between non-equilibrium, rapidly fluctuating C_2 . To test the impact of ML encoding on the ability to identify similarities between C_2 , we provided a target image and used Euclidean distance in both the ML latent space and raw input space vectors to suggest the top 300 most similar images to the target. The 300 selections from each method were shown to a panel of XPCS experts, who were each asked to determine how many of the suggestions are actually similar to the target (Figure 8, below). ML suggestion accuracy is higher than that of suggestions from the baseline across the board, and we note that (at least for this target image) ML suggestions have greater than 50% accuracy for the entire range of 300 C_2 . We also note the high variability among expert evaluations –

the difficulty of qualitative interpretation even for human experts highlight the need for a reliable automate approach.

Figure R8. Copy of Supplemental Figure 6, showing difference in clustering and UMAP embedding of ML latent space and raw data.

Figure R9. Copy of Supplemental Figure 7, comparing the accuracy of ML and raw data similarity evaluations as evaluated by experts.

Figure R10. Copy of Supplemental Figure 8, showing the distribution of nearest neighbor distances in the ML latent space and baseline raw data analysis.

We attribute this high accuracy and able to recognize similarities between data to sparsity in the latent space – processing raw data requires operating in 256^2 dimensions, where differences between individual data points are harder to discern compared to the reduced 64-D latent space. The well-known curse of dimensionality explains that as the dimension of a dataset increases, the distance between samples, and therefore the ability to distinguish subtle differences, decreases. Figure R10 below compares the distributions of 1st, 10th, and 100th nearest neighbor distance between all samples for the 64-dimensional latent space and 256^2 -dimensional baseline analysis. In this case, the separation of nearest neighbor distances signifies the ability of each model to distinguish between unique C_2 . ML neighbors show significantly wider separation than analysis on the raw data, signifying that reduced sparsity in the latent space helps the model to compare data.

We have included Supplemental Figure 5 (Figure 10, copied below) to address the issue of cherry-picking good results: indeed, for data which has less common or clear features the suggested nearest neighbors are not identical, however they are still similar enough that we humans can see visual relationships between the data.

Figure R11. Copy of Supplemental Figure 5. Randomly selected target C2, and nearest neighbors identified in the latent space. In this case, compared to Figure 4 of the main text, the neighboring samples are less similar. Still main features of the target C2 are identified.

Changes in manuscript:

We have added discussion of ML model selection for Section II, B and Supplemental Figure 5-8 (copied above as Figures R). Additionally, we have addressed our tests using simpler methods and the advantages/drawbacks of complicated machine learning models for scientific data in the Section II,C.

“Even though application of machine learning offers an opportunity to overcome bottlenecks in experimental data analysis, reliability, generalization to new data, and the danger of applying complex algorithms as a black box are areas of major concern for application of machine learning in scientific data. Additionally, the inner workings and results of the model can be difficult to interpret. With this in mind, ML results should be compared against analyses possible through simpler methods to evaluate the trade-off between increased accuracy at the cost of interpretability. To address this issue, we have performed additional tests to understand the benefits brought by studying our system in the ML-encoded latent space, compared to performing similar analyses on the raw data. For the following analyses, shown in Supplemental Figures 7-9), the ML-encoding refers to the 64-parameter latent space vector describing C2, while the raw data representation refers to simply flattening the 256 x 256-pixel images to vectors of size 256^2 such that both representations can be processed in similar ways. We find that clustering results and the appearance of the UMAP visualization are very similar regardless of the data representation C2, and therefore do not benefit from latent space analysis (Supplemental Figure 6).

This can be attributed to the fact that the main feature of the data is the mean intensity of C2 corresponding to the average dynamics, and that clustering algorithms groups data by this dominant feature regardless of the dimensionality of the data set. That being said, ML-encoding shines in identifying similarities and differences between data, which is seen as the distance between C2 in the raw and ML spaces rather than the assigned cluster label. To test the accuracy of ML-encoding versus analysis on raw image data, we manually identified a visually striking target image and found the 300 most similar C2 from a large dataset (13,000 unique C2) using euclidean distance in raw image space, and the ML latent space. XPCS beamline scientists with expertise analyzing XPCS data across soft- and hard-matter physics were asked to evaluate where the suggested C2 were in fact similar to the target. As shown in Supplemental Figure 7, ML suggestions achieved greater than 70% accuracy over the first 200 suggested images, while raw data analysis only saw comparable accuracy on the 20 nearest neighbors. The differences in expert evaluations highlights the need for automated methods - this task is difficult and subjective even for trained experts, so automating data processing removes bias and variability. The drastic difference in performance can be attributed to data sparsity, or the *curse of dimensionality* which states that distances between data points becoming increasingly indistinguishable as the dimensionality of the data increases. This is shown in Supplemental Figure 8, where we compare the distributions of first, 10th, and 100th neighbor distances between all C2 within both the raw data and the ML encoding. Ideally, the representation of the data should account for significant differences between near (similar) and far (dissimilar) neighbors. In Supplemental Figure 8 we see significant overlap between the first and 100th neighbor distance distributions indicating that similar and dissimilar patterns are indistinguishable. Therefore, distance between raw data points is a poor metric for capturing similarities and trends in the data. In contrast, there is more separation between the first, 10th, and 100th neighbor distances in the ML latent space, suggesting that the richer, learned representation from the ML model is more capable of describing subtle features of C2. In summary, the ability of the latent space to represent complex relationships between data in a simpler fashion enables easier and accelerated exploration of data sets which will be crucial for experiments at next generation light sources where the data production rates will make manual analysis impossible.”

=====

Referee comment:

In addition, the paper has some minor weaknesses:

(1) The paper tends to use the terms “class/classification” and “cluster” interchangeably, though they are generally considered to be distinct concepts in the ML literature. For example, the clusters found by k-means should be called “Cluster 0” etc, not “Class 0”.

Author response:

Thank you for pointing this out, we have adjusted the text accordingly

Changes in manuscript:

We have change wording in the manuscript to use ‘Cluster’ instead of ‘Class’

=====

Referee comment:

(2) Figure 1(D) is described as “denoised”, but there is no information presented to confirm that there is indeed noise in the cross-correlation visualisations.

Author response:

We see that this is unclear and that noise may not be the right term. We have adjusted the text and used the word smoothed instead.

=====

Referee comment:

(3) Figure 2 is described as showing clusters, but Classes 3 and 4 overlap, while Class 4 contains 3 completely separate groups.

Author response:

Clustering is performed and calculated in the (64-dimensional) latent space, while the visualization shows the UMAP projection onto 2D space. Therefore, while Cluster 4 appears to have clear sub-clusters in UMAP space, datapoints in these sub-clusters appear much closer together in the full latent space. UMAP projections can be used only qualitatively to observe the structure of the dataset, but exact positions and directions in this space are meaningless. In this light, it is not surprising that classes overlap for two reasons: 1. Since we see a nearly continuous transition in the data from rapid relaxation to stationary dynamics, the data does not fall into clear clusters and so “fuzzy” boundaries are expected, 2. If points at the boundaries are similar in the latent space they will appear similar/overlap in the UMAP projection.

Changes in manuscript:

We have expanded our discussion of UMAP visualization in Section II, B.

“Due to the high dimensionality of even the latent representation, further embedding is required to visualize the distribution of the encoded data and the clustering results. We used Uniform Manifold Approximation and Projection (UMAP) to transform the latent space of the dataset into two dimensions; this visualization is shown in Figure 2.

UMAP is closely related to t-distributed Stochastic Neighbor Embedding (tSNE), a more common method for non-linear dimensionality reduction. Both of these methods consider the local structure of the data distribution and attempt to project data points onto a lower dimensional manifold, however, in comparison to tSNE, UMAP distorts the data distribution such that it is uniformly distributed in the projection space. This helps maintain the global structure of the dataset, and generates projections which are more stable against variation in initialization and hyperparameters than those generated by tSNE. This visualization allows us to qualitatively check the accuracy of the clustering results by seeing whether optimal cluster centers coincide with the densest regions of the UMAP embedding.

Viewing images from each cluster (Figure 2) shows that relaxation times decrease with increasing cluster label. Following the trajectory across the UMAP distribution in Figure 2A from Cluster 0 to Cluster 4, we can see the transition between nearly stationary dynamics in Cluster 0 (high correlation across long times relates to slow structural changes) to slow evolution in Cluster 1 (seen as flat C2 features with lower intensity), to increasingly fast evolution in Clusters 2, 3, and 4. Distances in these embedding spaces should not be quantitatively compared, and only serve a qualitative metric of similarity between data points. Outlying yellow regions assigned to Cluster 4 in Figure 2A appear very similar to the sample image from Cluster 4 in Figure 2B - they have rapid decay and very low intensity away from the main diagonal. Finally, we note the nearly continuous transition between clusters in the latent space which is evidence that the data set contains a continuous spectrum of relaxation dynamics.”

=====

Referee comment:

(4) In Fig 3, it would be helpful to arrange the two figures with A on top of B, not to the left of B, so that the reader could see more easily how they align versus time.

(5) The discussion of Fig 3 is made difficult because of references to “green” and “dark green” bands, when in fact there are 3 shades of green and a never-mentioned blue colour.

Author response:

We have arranged plots in Figure 3 vertically, and changed the text to avoid discussion of colors, and focus on the distribution of clusters.

Changes in manuscript:

“The growing peaks of slow dynamics (low cluster numbers) seen between shear stress minima indicate that tracking material behavior through latent space encoding is able to tie intermittent microscopic dynamics to macroscopic rheological changes.”

=====

Referee comment:

(6) The discussion at the end of Section II is quite speculative in nature, with little evidence to support it.

Author response:

We have rewritten the discussion to make it more clear, and to focus on specific improvements enabled by our model and next steps for research. In Section II,C we have highlighted specifically how our approach can be used to guide model selection and aid interpreting non-equilibrium dynamics, and discussed how future work could use our ML-based approach to address challenges in characterizing micro-rheology. In the discussion section we have mentioned commented on the presence of dynamics at different length and timescales in out-of-equilibrium materials to demonstrate how our method of extract statistical properties of large datasets can aid in data interpretation, and to highlight the relevance of this approach across a variety of materials.

Changes in manuscript:

Our discussion in both Section II and III have been extensively rewritten to be more concrete and provide tangible examples of how our method can aid scientific discovery. We have adjusted the text to focus explicit ways that our workflow addresses gaps in current methods for analyzing XPCS data.

Section II,D:

“Results from analysis in Figures 3A-D suggest that: 1. Dynamics slow down through time, initiated in the flow direction, 2. the intermittent microscopic dynamics are tied to macroscopic rheological fluctuations (however, not all microstructural changes are reflected in the macroscopic response), and 3. in addition to the rate of dynamics, the appearance of C2 differs drastically depending on scattering angle.

Research studying dynamics in materials generally aims to define a characteristic time scale which describes the evolution mechanism. While XPCS is capable of precisely capturing material dynamics by measuring the decorrelation time between successive scattering frames, a physical model of the process is still required to extract quantitative information describing, for example, particle motion or relaxation rate in the sample. Model selection is difficult for non-equilibrium systems, where either physical models do not exist, or the selection of a model is complicated by the presence of rapid unexpected changes in dynamics. Furthermore, most theoretical frameworks describe the mean behavior of a system, yet the nature of XPCS experiments probes local dynamics which may differ from expected behavior on experimental time scales, selection of a model based on visual inspection of a few sample C2 from an entire experiment is difficult and potentially unreliable. Therefore, the ability to categorize different types of XPCS C2 patterns to guide the development of structural and dynamics models for evolving materials is crucial.

Considering the conclusions drawn from analysis of Figure 3 guides us towards the selection of the heterodyne scattering model to describe our data, which occurs when a difference in velocity between two or more dynamically distinct components in the system leads to constructive/destructive interference in correlation patterns, and enables the extraction of system-independent information about the relative velocity between components. Heterodyne scattering produces a uniquely recognizable fringe pattern in XPCS C2 in specific scattering directions. This phenomenon has been documented in some experimental systems, however it has not been observed in experiments of colloidal glasses. Thus, our AI-guided model selection provides unique insight into microstructure evolution in complex materials, and enables future research to quantify the properties and dynamics in this system.”

“XPCS datasets may contain thousands of unique C2, making it infeasible to comb through all the data to identify instances with similar characteristics. One can imagine an extension of our automated approach where the user defines a C2 topology of interest and defines a distance from the sample C2 to create a neighborhood of C2 which are likely to look similar. After identifying closely-related C2, corresponding measurements showing experimental time, shear stress, shear rate, viscosity, etc., along with metadata describing acquisition parameters, e.g. scattering vector as shown in Figure 3 and Supplemental Figure 4, or beam position on the sample, could be aggregated to understand the conditions in which the target behavior occurs. Observing the distribution of system properties throughout the latent space will allow scientists to directly relate C2 appearance to physical properties in a general way which can be applied to any *in situ* XPCS experiment.”

Section III:

“Our work has focused on encoding and categorizing full C2, however out-of-equilibrium systems exhibit multiple timescales, both within a single C2, representing the milliseconds to seconds timescale, and

across many C2 in an entire experiment. Moreover, visual inspection of experimental non-equilibrium C2 show that dynamics in our glassy colloidal system are intermittent in many cases- rather than constantly changing dynamic characteristics, transitions appear as unpredictable changes between otherwise near-equilibrium states. Aside from temporal heterogeneity, spatial heterogeneity further complicates understanding non-equilibrium processes since comparison of bulk scale measurements with XPCS measurements on a small area of the sample requires mechanistic descriptions which can cross length scales. In light of these challenges, understanding such transitions requires a systematic and unbiased method for observing and recognizing anomalous changes in large sets of data. Machine learning methods are an ideal choice for capturing subtle transitions while removing human bias, and our method represents a step towards automatic recognition of fluctuations in time-resolved x-ray scattering, and understanding how these fluctuations relate to measurable properties. Future work could extend our method to not only cluster and track dynamics between C2, but within sections cropped out of individual C2 to investigate how relaxation behavior changes as a function of timescale. Previous studies have demonstrated the fractal nature of relaxation processes in colloidal gels, meaning that structure and dynamics vary in a self-similar fashion across length and time scales. Our ability to track these changes, both within and across C2, with statistical certainty will allow greater understanding of how and why these mechanisms occur in disordered materials. Interestingly, researchers have also identified fractal relationships between structure and dynamics in metallic glasses. As our approach is not specific to studying colloidal suspensions and can be adapted for other classes of materials simply by retaining the CNN, this is another interesting case where our AI-NERD approach could be applied to understand relaxation across length and time scales.”

=====

Referee comment:

(7) Reference 63 is incomplete.

Author response:

Thank you for pointing this out, we have fixed the reference.

=====

Referee comment:

A final observation: given that the data is continuous in UMAP space, and is not really arranged into discrete clusters, I think it would be more interesting to fit a regression function to the latent space data, rather than clustering it.

Author response:

We thank the reviewer for this insightful comment, and agree that regression may be better suited, in some regards, to describing the continuous nature of the material evolution. That being said, we have chosen to describe the data in terms of discrete clusters to ease visualization and immediate interpretation. For example, in Figure 3B (see Figure R7 above) the relatively low number of cluster values (5 clusters) compared to the number of available measurements in each time-point (18 C2 at each time) allows us to build a histogram tracking dynamic changes with an appropriate “bin width” for interpretation compared to a regression case where all C2 would have a unique value making visualization of the distribution difficult. Additionally, we find that significant changes in C2 visual

appearance occur while traversing the latent space perpendicular to the main axis as evidenced by Figure 3D (Figure R7 above) and Supplemental Figure 4, which implies that both C2 appearance and material properties vary within small regions of the latent space. Finally, while UMAP has the advantage of stability with respect to random initialization over the tSNE embedding algorithm, the appearance of the embedding manifold is still based on a randomized optimization process. We expect that UMAP results are reproducible in terms of trends and general relationships within the data, however any regression analysis may be misleading since the exact embedding manifold can change with initialization.

We feel that while regression analysis may be complementary to our technique, reducing the data to a single line loses more information about the system than clustering.

REVIEWER COMMENTS

Reviewer #1 (Remarks to the Author):

The authors have addressed my questions and concerns. I have no further comments.

Reviewer #2 (Remarks to the Author):

I appreciate the major revision of the manuscript which helped to improve the paper considerably. The new approach presented in the paper is very interesting, timely and a much needed step forward for dealing with the data avalanche in the xpcs community.

I still wished for a better and deeper connection to physics ongoing in the sample - but I also realize that this is not in the time-window of the current paper. But definitely this paper lays the ground-work for achieving the next step of physics informed AI based xpcs analysis. So I suggest the editors to accept the paper for publication.

A few more technical details/comments below:

Page 5:6

In order to assess the effectiveness of the methodology, they employed a pre-trained model to produce data with added noise. The lack of Information about the Trained Model and Its Training Data: Without knowledge of the trained model, its trustworthiness, and the nature of its training data, it's challenging to assess the validity of the approach.

Standard Deviation (σ) of 0.3 for Testing Accuracy: The choice of $\sigma = 0.3$ for adding Gaussian noise is not inherently problematic from a mathematical or statistical perspective. However, it may not be a suitable test for assessing the accuracy or robustness of the model in all cases. If the goal is to evaluate the model's performance under conditions of moderate noise ($\sigma = 0.3$), it can be a relevant test. If the goal is to evaluate the model's robustness to higher levels of noise, extreme scenarios, or other types of noise, then additional tests with different levels of noise or perturbations may be necessary.

Page 6:7

"We note that a major concern in the field of representation learning is the development of ML models which enforce continuity and orthogonality in the latent space to produce latent dimensions which are independent and may directly relate to physical features. While these approaches have proven successful in the x-ray characterization papers mentioned above, we have found that using such regularization schemes did not allow us to produce accurate reconstruction of C2 during the training process. We have chosen to use a standard convolutional autoencoder, in contrast to a variational autoencoder (VAE) or other type of model, to maximize the amount of information our model can learn at the expense of the guarantee of a continuous latent space."

The text is challenging to understand. Providing a bit more detail about why regularization schemes didn't work for this particular problem would improve the text.

Page 7

"B. C2 Clustering and Latent Space Analysis"

The statistical justification for using the k-means algorithm for clustering should be clarified, particularly in terms of the assumptions that underlie its effectiveness (i.e. Uniform distribution of data points, equal variance in all dimensions, and uniform prior probability for finding number of

clusters(k)).

K-mean with elbow method:

k-means algorithm is known to be sensitive to outliers since the computing mean is influenced by extreme data points.

The choice of the "elbow point" or the point where the rate of decrease in the sum of squared distances (SSD) starts to slow down can be somewhat subjective. Different individuals might interpret the plot differently and select different values of k.

The elbow method can help identify a reasonable choice for k, but it does not guarantee that the clustering solution obtained with that value of k is the globally optimal solution. It only provides a heuristic for selecting k.

The elbow method is based solely on the within-cluster variation (SSD) and does not take into account the density or spread of data points within clusters.

The elbow method assumes that clusters are convex, roughly spherical, and well-separated. In cases where clusters have complex shapes, overlapping regions, or varying densities, the elbow method may not provide a clear and accurate choice for the number of clusters.

Page 8

"C. Probing non-equilibrium dynamics using trained ML model"

The results presented here are based on the assumption of a 5-cluster system, determined through the application of the Elbow method. However, it is essential to test this outcome, either through the utilization of an alternative clustering algorithm for validation or by implementing a clustering algorithm designed to handle an unknown number of clusters. Moreover, I strongly advocate for conducting a comparative analysis of various clustering algorithms.

Reviewer #3 (Remarks to the Author):

I thank the authors for carefully considering my review and for the clarity with which they have presented the updates they made in response to my comments.

While the paper is substantially improved, I still have some concerns about it, as enumerated below.

1: In my initial review, I noted that the novelty of the work is unclear to me, since it is well understood how to use encoders-decoders to generate lower-dimensional latent-space representations of high-dimensional data. The authors did not respond to this comment. For a journal with a broad audience such as Nature Communications, I think it is important to have clearly articulated novelty.

2: As I noted before, in the original paper, the authors did not present any empirical evaluation to demonstrate that their technique is better than alternatives from the literature. This still has not been done.

3: The authors have responded to my comment related to the above one, by presenting a comparison versus a trivial nearest-neighbour approach. However, as I summarise below, the new comparison still has substantial problems.

3a: The authors use the term "ML" and "ML encoding" which is confusingly imprecise; why not use the name AI-NERD if you are referring to your own method? There are many different ML methods, and k-nearest neighbours is considered an ML algorithm also.

3b: There authors claim, with reference to Supplemental Figure 7: "ML suggestions achieved greater than 70% accuracy over the first 200 suggested images, while raw data analysis only saw comparable accuracy on the 20 nearest neighbors." I have studied the figure carefully and I cannot see any support for this claim. If I look at the solid green line representing the expert who

gives the most 'generous' scores, I see that 70% accuracy corresponds to about 150 suggested images, not 200. The authors should clearly explain in the text what data they are looking at to draw this conclusion.

3c: The whole assessment by experts is questionable, since no information is provided about how the experts were selected (were they the authors or the authors' colleagues, who might be implicitly biased in favour of positive results for the CNN approach?) and how the evaluation was conducted (was it a blind review or did experts know which set of results they were reviewing?)

3d: The authors state, with reference to Supplemental Figure 8, speaking about the raw data: "we see significant overlap between the first and 100th neighbor distance distributions...". About the encoded data, they say: "there is more separation between the first, 10th, and 100th neighbor distances in the ML latent space". Again, I have studied the figure carefully, and while I see that the raw data distributions are narrower than the CNN ones, the percentage overlap does not appear to be much different.

3e: In this and in other parts of the paper, rather than just making subjective claims with reference to figures, the authors should investigate how to statistically verify the significance of the separation between distributions and the statistical significance of their claims.

3f: Related also to Supplemental Figure 8, there is an unexplained high value at the start of the CNN 100th NN graph. What is the explanation for this?

Reviewer #1 (Remarks to the Author):

The authors have addressed my questions and concerns. I have no further comments.
We thank the reviewer for their assessment, and their input in the previous revision.

Reviewer #2 (Remarks to the Author):

I appreciate the major revision of the manuscript which helped to improve the paper considerably. The new approach presented in the paper is very interesting, timely and a much needed step forward for dealing with the data avalanche in the xpcs community.

I still wished for a better and deeper connection to physics ongoing in the sample - but I also realize that this is not in the time-window of the current paper. But definitely this paper lays the ground-work for achieving the next step of physics informed AI based xpcs analysis. So I suggest the editors to accept the paper for publication.

We thank the reviewer for their comments and input. We understand the reviewer's concern about the depth of our connection to physics. Indeed, as the reviewer points out, our goal is to lay the groundwork for further physics-based analysis of XPCS data.

A few more technical details/comments below:

Page 5:6

In order to assess the effectiveness of the methodology, they employed a pre-trained model to produce data with added noise. The lack of Information about the Trained Model and Its Training Data: Without knowledge of the trained model, its trustworthiness, and the nature of its training data, it's challenging to assess the validity of the approach.

We apologize for this confusion. The pre-trained model simply refers to the AI-NERD model described in the paper, applied to generating synthetic data after training. The trustworthiness and validity of the approach has been supported by data presented in Supplementary Figure 1.

Changes to the manuscript:

- **Modified description of the generative task to say: "To ensure that the latent space encoding accurately represents the distribution of experimental data, after training we used the model to generate artificial C2 by adding noise to the latent representation of real C2."**
- **Removed other references to the train/pre-trained model to avoid confusion**

Standard Deviation (σ) of 0.3 for Testing Accuracy: The choice of $\sigma = 0.3$ for adding Gaussian noise is not inherently problematic from a mathematical or statistical perspective. However, it may not be a suitable test for assessing the accuracy or robustness of the model in all cases. If the goal is to evaluate the model's performance under conditions of moderate noise ($\sigma = 0.3$), it can be a relevant test. If the goal is to evaluate the model's robustness to higher levels of noise, extreme scenarios, or other types of noise, then additional tests with different levels of noise or perturbations may be necessary.

We thank the reviewer for this pertinent observation. While the choice of σ can have a large impact on the model's ability to generate new samples reliably, we note that our goal is not to use AI-NERD as a generative model. These tests were designed to prove that regions in the latent space do in fact correspond to similar C2 features even in the absence of latent space continuity and conditioning. As our goal in AI-NERD analysis is to gauge similarity between real experimental C2 via Euclidean distance in the latent space, we believe that tests in the moderate noise regime are sufficient to show that neighborhoods in the latent space contain similar data.

Changes to the manuscript:

- Adjusted the discussion of choice of sigma in the generative tests to highlight the purpose of these tests and that our model is not intended for use as a generative model: " $\sigma = 0.3$ represents the average standard deviation of all latent variables for the encoded test dataset. We note that, while generating synthetic data from real latent representations is an effective way to ensure that the latent space properly captures the distribution of real data, our goal is not to use AI-NERD as a generative model but rather to explore the distribution of experimental data in a reduced space. Therefore, the moderate noise condition of $\sigma = 0.3$ is sufficient to verify that small regions in the latent space contain similar real data. "

Page 6:7

"We note that a major concern in the field of representation learning is the development of ML models which enforce continuity and orthogonality in the latent space to produce latent dimensions which are independent and may directly relate to physical features. While these approaches have proven successful in the x-ray characterization papers mentioned above, we have found that using such regularization schemes did not allow us to produce accurate reconstruction of C2 during the training process. We have chosen to use a standard convolutional autoencoder, in contrast to a variational autoencoder (VAE) or other type of model, to maximize the amount of information our model can learn at the expense of the guarantee of a continuous latent space."

The text is challenging to understand. Providing a bit more detail about why regularization schemes didn't work for this particular problem would improve the text.

We thank the reviewer for pointing this out. We have modified the text to make this more clear. To do so, we discuss exactly how more complicated models such as VAE differ from our model during the training process, and we address our expectations for why this approach is not effective for our data.

Changes to the manuscript:

"The development of autoencoder models which enforce continuity and orthogonality in the latent space, such that latent dimensions are independent and hopefully interpretable, is a major concern in the field of representation learning. A common approach for latent space conditioning is seen in variational autoencoders (VAE), where an additional loss term is used in the training process to enforce that latent variables are drawn from multivariate gaussian distributions. Enforcing the shape of latent variable distributions helps to develop continuity in the latent space. Further loss or regularization terms can be added to neural network model to enforce orthogonality between latent parameters. While these additional constraints produce learned representations which are often more directly interpretable, the model training process is significantly more difficult and optimizing these conditioning factors comes at the cost of sacrificing image/data reconstruction quality. For instance, a β -Autoencoder attempts to address this concern by using a parameter β to weight the relative importance

of latent space conditioning and reconstruction error, however choice of β can be difficult and depends on the exact goal of the ML task. While these approaches have proven successful in the x-ray characterization papers mentioned above, we have found that using a VAE framework drastically deteriorated the quality of output reconstructions such that we could not trust that latent representations corresponded with input data. We attribute the model's inability to accurately represent our data in the presence of latent space training constraints to the large variability in our training set which may make distillation of the data into a fundamental set of parameters difficult. Therefore, we have chosen to use a standard convolutional autoencoder to maximize the amount of information our model can learn at the expense of the guarantee of a continuous latent space."

Page 7

"B. C2 Clustering and Latent Space Analysis"

The statistical justification for using the k-means algorithm for clustering should be clarified, particularly in terms of the assumptions that underlie its effectiveness (i.e. Uniform distribution of data points, equal variance in all dimensions, and uniform prior probability for finding number of clusters(k)).

K-mean with elbow method:

k-means algorithm is known to be sensitive to outliers since the computing mean is influenced by extreme data points.

The choice of the "elbow point" or the point where the rate of decrease in the sum of squared distances (SSD) starts to slow down can be somewhat subjective. Different individuals might interpret the plot differently and select different values of k.

The elbow method can help identify a reasonable choice for k, but it does not guarantee that the clustering solution obtained with that value of k is the globally optimal solution. It only provides a heuristic for selecting k.

The elbow method is based solely on the within-cluster variation (SSD) and does not take into account the density or spread of data points within clusters.

The elbow method assumes that clusters are convex, roughly spherical, and well-separated. In cases where clusters have complex shapes, overlapping regions, or varying densities, the elbow method may not provide a clear and accurate choice for the number of clusters.

We agree that the elbow method, as applied both for kmeans clustering inertia and the Silhouette score, can be subjective. Since our analysis is meant to be qualitative and designed only to show trends of dynamic evolution rather than specific features of individual clusters, the choice of cluster number does not significantly impact our analysis. To demonstrate this, we have performed similar clustering and cluster evolution analysis using three, five (as in the main text), and eight clusters in Figure R1 below – while specific clusters to which each data point is assigned may change, the evolution shows the same trends and allows us to draw the same conclusions: that microscale dynamics clearly correlate with macroscopic changes and that dynamics have an angular dependence. We acknowledge, however, that this will not be the case for all datasets and the choice of cluster number may be very important in other applications.

We have added a statement to the methods section to discuss the choice of clustering algorithm and number, and the impact on resulting analysis.

In addition to changes in the manuscript, we have included all code needed to perform systematic tests on clustering algorithms and number choice, along with corresponding figures, in the github repository associated with this code so that future users will be able to objectively compare algorithms without any barrier to set up.

Figure R1. Comparison of analysis results with varied cluster number.

Changes to Manuscript:

- The following text has been added to our description of the clustering process in the methods section: “In this work we have chosen to use the K-means clustering algorithm because of its scalability to large datasets which enables rapid analysis and iteration. However, the choice of clustering algorithm must be made with regards to its underlying assumptions, and we note that the use of AI-NERD in other applications which show strong structuring, non-uniformity, or large numbers of outlying data points, the K-means algorithm may not be most effective. We performed systematic tests of other clustering algorithms, especially those which do not require user-specified number of clusters and found that these results in general converge to an unrealistic number of clusters and depend heavily on hyperparameter tuning. Aside from the choice of clustering algorithm, the choice of the number of clusters has potential to impact ensuing analysis. While the elbow method is commonly applied to determine the ideal number of clusters, the exact placement of the elbow can be subjective. We have found that our clusters correspond to the relative rate of dynamics in a nearly continuous distribution, and therefore variations in the number of clusters does not significantly impact our findings. We caution that in other applications where stronger clustering behavior is present, careful attention must be paid to clustering results and visualization. Plots showing our systematic tests of clustering algorithms and cluster number choices can be found in our analysis code.”

“C. Probing non-equilibrium dynamics using trained ML model”

The results presented here are based on the assumption of a 5-cluster system, determined through the application of the Elbow method. However, it is essential to test this outcome, either through the utilization of an alternative clustering algorithm for validation or by implementing a clustering algorithm designed to handle an unknown number of clusters. Moreover, I strongly advocate for conducting a comparative analysis of various clustering algorithms.

As mentioned in response to the previous point, we have demonstrated that our qualitative analysis of dynamic evolution does not change as a function of cluster number.

We have conducted tests using other clustering algorithms to understand how the underlying assumptions of each algorithm can impact the identification of clusters in the dataset. Figure R2 shows the clustering results from all clustering algorithms available in the scikit-learn package which do not require a user-specified cluster number. Aside from the BIRCH algorithm which, to the best of our ability, shows three clusters regardless of parameter choice, these results show that clustering algorithms which do not require a user-specified number of clusters are not able to identify clusters in our dataset – algorithms tend to suggest clusters with very few data points, and assign the majority of data to a single cluster. The exact reason for this is difficult to diagnose since it is impossible to visualize the 64-D latent space and the UMAP projection is only an approximation of the global structure, but we attribute these issues to the vast range of C2 appearances in the dataset not falling into distinct clusters.

We have included the results of our comprehensive comparison of clustering algorithms in github repository associated with this manuscript and have included a statement in the text to warn future users that the choice of clustering algorithms can vary by use case.

Figure R2. Comparison of scikit-learn clustering algorithms without manual selection of cluster number.

Reviewer #3 (Remarks to the Author):

I thank the authors for carefully considering my review and for the clarity with which they have presented the updates they made in response to my comments.

While the paper is substantially improved, I still have some concerns about it, as enumerated below.

1: In my initial review, I noted that the novelty of the work is unclear to me, since it is well understood how to use encoders-decoders to generate lower-dimensional latent-space representations of high-dimensional data. The authors did not respond to this comment. For a journal with a broad audience such as Nature Communications, I think it is important to have clearly articulated novelty.

We thank the reviewer for pointing this out. Encoder-decoder architectures and representation learning are indeed widely applied. Previous applications in the realm of materials characterization, as noted by reviewer #1 in the previous round of revisions, have focused on characterizing physical/chemical properties of materials using more common high-throughput experiments. The novelty of our work lies, first, in the application of these techniques in XPCS experiments which is a characterization method with very unique and unmatched capabilities. With more and more fourth generation synchrotron sources coming online, the data output using a brighter and more coherent x-rays will yield a significant increase of XPCS image data. The AI-driven method outlined in this manuscript will play a significant role in meeting the demands of increased data production rates, even opening the possibility of data processing on-the-fly. Secondly, the use of latent representation learning to understand time-dependent non-equilibrium processes provides a new way to analyze dynamics data where conventional analysis methods may be impossible, ambiguous, or unreliable. We have modified the manuscript to clearly state the novelty of our research in the introduction, and to clearly address the differences between our approach and those of others later in the text.

Changes in the manuscript:

- Added to the introduction: In comparison to other applications of representation learning to understand materials systems, our method is the first to use unsupervised learning in time-resolved experiments to guide understanding of non-equilibrium dynamics without requiring background information on the sample or experimental setup. Rapid changes in non-equilibrium dynamics are invaluable for understanding how materials respond to stimuli *in situ* across wide ranges in space and time, however capturing these fluctuations is difficult with experiments outside of time-resolved coherent x-ray scattering techniques such as XPCS. Increased x-ray flux and coherence at next-generation light sources will increase signal in XPCS correlation calculations and push analysis towards the richer C2 representation of dynamics compared to the traditional g2. Therefore, the need for reliable and general methods for analyzing complex C2 is greater than ever. Our work demonstrates the capability of applied machine learning to accelerate scientific discovery through x-ray characterization, and to move towards the ability to fully utilize experimental capabilities for high-frame rate, high-flux/coherence experiments available at next-generation synchrotron light sources.
- Added to the discussion section: "Machine learning methods are an ideal choice for capturing subtle transitions while removing human bias, however to date applications of machine learning to scientific data focus on experimental techniques where data have clearly discernable features, e.g. peaks with finite positions and widths. In contrast, our method focuses on using AI

to identify important features in data which are difficult or ambiguous to interpret by eye, even for human experts.”

2: As I noted before, in the original paper, the authors did not present any empirical evaluation to demonstrate that their technique is better than alternatives from the literature. This still has not been done.

The main objective of our work is not to demonstrate a novel machine learning model, and therefore our focus is not to argue that the specific CNN used in this manuscript is better than other architectures. Instead, we are demonstrating the use of autoencoders/representation learning in a completely new application space: high-data rate x-ray photon correlation spectroscopy (XPCS). With the improvement of x-ray coherence and brightness of synchrotron sources, XPCS is becoming an important tool for studying non-equilibrium dynamics in many systems. Typically, XPCS analysis requires exploration and reduction of large and complicated datasets. Thus, the AI-NERD model presented here has the ability to quickly identify new physics in non-equilibrium systems, and expand the utility of a relatively-untapped experimental modality. We focus on latent representations to elucidate subtle changes in time-resolved datasets, which has not been accomplished in previous publications.

We acknowledge that autoencoders are widely used in the literature, however specific models are not interchangeable without modification for cases of domain adaptation. For example, while the work by Routh or Grossutti (references 53 and 54 in the text) also use autoencoders to study materials, both focus on one-dimensional datasets – adapting these models for image data would require significant architectural changes and an exponential increase in the number of trainable parameters. Even in comparison to Konstantinova or Timmerman (references 46-48) who also use autoencoders to study XPCS C2, modifications to the number of convolutional layers/filters per layer and hyperparameters would be required for application on our datasets; these studies focus on extremely noisy C2 and nearly-equilibrium/slowly evolving C2, respectively, which are significantly different and less complex than the data encountered in our work.

In all these approaches including our own, the goal of the research is not to present a single autoencoder model with optimal performance on physical datasets but is rather to demonstrate the use of a flexible unsupervised machine learning framework to uncover various physical phenomena. Therefore, rather than focusing on adaptation and optimization of autoencoder models (which is addressed in the methods section and Supplemental Figure 8-9), we have focused on comparing our AI-driven analysis to a more conventional approach of XPCS data analysis which has major limitations as well laid out in the manuscript.

3: The authors have responded to my comment related to the above one, by presenting a comparison versus a trivial nearest-neighbour approach. However, as I summarise below, the new comparison still has substantial problems.

3a: The authors use the term “ML” and “ML encoding” which is confusingly imprecise; why not use the name AI-NERD if you are referring to your own method? There are many different ML methods, and k-nearest neighbours is considered an ML algorithm also.

We appreciate the referee pointing out this confusion, and have adapted figures and text in the results and discussion sections, using “AI-NERD” to refer to our method instead of ML/ML-encoding.

Changes to manuscript:

- Throughout the section *Probing non-equilibrium dynamics using AI-NERD*, references to ML were replaced with AI-NERD

3b: There authors claim, with reference to Supplemental Figure 7: “ML suggestions achieved greater than 70% accuracy over the first 200 suggested images, while raw data analysis only saw comparable accuracy on the 20 nearest neighbors.” I have studied the figure carefully and I cannot see any support for this claim. If I look at the solid green line representing the expert who gives the most ‘generous’ scores, I see that 70% accuracy corresponds to about 150 suggested images, not 200. The authors should clearly explain in the text what data they are looking at to draw this conclusion.

We thank the reviewer for pointing this out. It appears that we missed these statements when deciding whether to compare to the 50% vs 70% accuracy thresholds. In our new revision we state all accuracy claims relative to the 50% threshold, under the presumption that this signifies accuracy better than random chance. In addition to the existing black line representing 50% accuracy, we have adapted Supplemental Figure 7 to include plots showing the average expert evaluation on both datasets, and vertical lines showing when the average expert evaluations cross the 50% accuracy threshold to help guide the reader’s eye and clarify how we derive our claims of accuracy. Based on this, we see that on average AI-NERD suggestions achieve greater than 50% accuracy for 260 suggested C2, while the same accuracy is only found for the first 70 suggests using conventional analysis.

Changes to the manuscript:

- Supplemental Figure 7 has been modified to include plots showing the average expert evaluation on both datasets, and vertical lines showing when both datasets cross the 50% accuracy threshold
- Text has been modified that we are judging based on the average evaluation score, and provide explicit image numbers where each datasets crosses the 50% threshold: “As shown by the black curves in Supplemental Figure 7 which represent the average of all expert evaluations, AI-NERD suggestions achieved greater than 50% accuracy for 260 out of the 300 suggested images, while raw data analysis only saw comparable accuracy on the 70 nearest neighbors (shown as vertical black lines in Supplemental Figure 7)”

Supplemental Figure 7. A blind test to evaluate the accuracy of AI-NERD. Three XPCS experts were given a target C2 image and two unlabeled datasets consisting of 300 unique C2, one set generated from AI-NERD and another set generated from Raw data based on the proximity to the target image in the latent space (see Supplementary Figure 6). The experts were asked to go through each set and identify which samples showed similar features to the given target C2. Suggestion accuracy is defined as the fraction of C2 which experts identified as similar to the target. Accuracy is plotted as a function of the number of C2 encountered in nearest-neighbor order to track how visual similarity changes with euclidean distance in the AI-NERD and raw data spaces.

3c: The whole assessment by experts is questionable, since no information is provided about how the experts were selected (were they the authors or the authors' colleagues, who might be implicitly biased in favour of positive results for the CNN approach?) and how the evaluation was conducted (was it a blind review or did experts know which set of results they were reviewing?)

Experts include the authors on this manuscript; however, the experiment described in Supplementary Figure 7 was conducted as a blind review. The experts were given a target C2 image and two unlabeled datasets consisting of 300 unique C2, one set generated from AI-NERD and another set generated from Raw data based on the proximity to the target image in the latent space (see Supplementary Figure 6). They were asked to go through each set and identify which samples showed similar features to the given target C2.

While we understand the concern that expert evaluation may be biased, especially if experts are colleagues, comparison with human performance is a cornerstone of evaluating AI models, particularly as a metric for AI model performance in cases which are difficult even for human experts (Batra, *et al.*, Nature Chemistry **14** 1427-1435 (2022)). Batra, *et al.*, highlight that aside from human bias in the sense mentioned by the referee, human experts' bias from previous experiments may limit their ability to recognize overarching features of data when they are focused on their intuition – avoiding this issue is exactly why reliable automated processing is required for understanding non-equilibrium dynamics in XPCS. Moreover, in many supervised machine learning tasks expert evaluation is required for

developing the ground truth training set on which the model relies. Therefore, since this was conducted as a blind review, we consider that evaluation by experts with decades of experience is analogous to determining the ground truth, rather than providing a biased evaluation.

Changes to manuscript:

- We have changed the phrasing used to describe the process of expert evaluation: “XPCS beamline scientists with decades of experience analyzing XPCS data across soft- and hard-matter physics were asked to evaluate where the suggested C2 were in fact similar to the target in a blind review process: each expert was handed two sets of 300 C2, without knowing how the sample C2 were obtained or labeled. We note that we provided the experts with deliberately vague instructions, asking whether C2 are similar to the target without defining what features of the target were important; in this way, humans were essentially provided with the same information as the AI-NERD model.”

3d: The authors state, with reference to Supplemental Figure 8, speaking about the raw data: “we see significant overlap between the first and 100th neighbor distance distributions...”. About the encoded data, they say: “there is more separation between the first, 10th, and 100th neighbor distances in the ML latent space”. Again, I have studied the figure carefully, and while I see that the raw data distributions are narrower than the CNN ones, the percentage overlap does not appear to be much different.

After carefully considering reviewer’s comment, we have determined that it is not possible to quantitatively compare distances metrics between spaces with differing dimensionality. Additionally, comparing distribution statistics such as peak widths, positions, or overlap as in Supplemental Figure 8 is susceptible to bias for the same reason. While it was not our intention, we feel that the data presented in Supplemental Figure 8 may be confusing to readers and have decided to remove this figure from the Supplemental Information.

The purpose of this figure was to suggest that the rich representations obtained from AI-NERD enabled more detailed comparison between data. As this claim is already supported by the expert analysis in Supplemental Figure 7 and by our response to point 3c, we feel that the removal of Supplemental Figure 8 does not impact our claims and conclusions and reduces confusion for a general reader.

Changes to manuscript:

- Removed Supplemental Figure 8 and the surrounding discussion in the manuscript.

3e: In this and in other parts of the paper, rather than just making subjective claims with reference to figures, the authors should investigate how to statistically verify the significance of the separation between distributions and the statistical significance of their claims.

We have gone through the entire paper to ensure that any speculative/subjective claims are removed, including Supplemental Figure 8.

Changes to manuscript

- Remove Supplemental Figure 8.

- Removed the following sentence from the manuscript: “The ability to understand spatial heterogeneity and its impact on rheology would have significant impact on our understanding of jamming and viscoelasticity in colloidal suspensions, especially given our discovery that multiple dynamics components exist within regions the size of the x-ray probe coherent volume”

3f: Related also to Supplemental Figure 8, there is an unexplained high value at the start of the CNN 100th NN graph. What is the explanation for this?

The large peak in the 100th nearest neighbor curve in Supplemental Figure 8 can be attributed to the CNN’s superior ability to find similarity between samples. While the data show widely varying behavior, there are a few types of C2 patterns which repeat frequently: near-equilibrium dynamics, and fringed heterodyne dynamics (as described in the main text). The essentially bi-modal distribution of the 100th nearest neighbor distances suggests that AI-NERD recognizes one subset of data where patterns repeat over 100 times, while the remainder of the data is more disparate. The fact that this peak is absent in the 1st and 10th nearest neighbor distributions can be attributed to a relatively even distribution of data in the latent space over short distances.

The large peak in 100th nearest neighbors found in AI-NERD evaluation is not seen in the 100th nearest neighbor distribution calculated through raw data analysis. As distances essentially represent the pixel-wise error between data, raw data analysis cannot identify similarity between data which shows, for example, fringed patterns with varying frequency.

REVIEWERS' COMMENTS

Reviewer #3 (Remarks to the Author):

A: I feel that every time I examine a result in this paper closely, I spot an error. This causes me significant concern, because I did not study every single result in the paper so closely, so I believe there might well be other problems lurking elsewhere. In particular:

A1: In my previous comment 3b, I pointed out that the authors' claims about Supplemental Figure 7 were incorrect. They respond by claiming that this was just a mistake in deciding whether to use 50% or 70% accuracy thresholds, but their original claim accuracy over the first 200 suggested images does not hold at either 50% or 70%. We can see that they have modified their claim now to "50% accuracy for 260 out of the 300 suggested images".

A2: In my previous comment 3d, I again questioned the authors' claims about what is shown in Supplemental Figure 8, and they have now responded by deleting the whole figure.

B: I also have significant concerns about the response to my comment 3c, in which I asked about the composition of the group of experts who performed "blind" assessments. Specifically:

B1: The authors provide a detailed justification for why we might need expert review; this is completely irrelevant, I am not arguing that expert review is never needed, I am simply requesting that the readers be provided with appropriate information about the review process.

B2: The authors acknowledge in their response that the experts providing blind assessments include the authors. However, they do not include that detail in the paper. Why are the authors concealing this fact in the publication? Is it because readers will recognise it as a conflict of interest?

B3: The authors still do not provide any details about how they ensure that the review is truly blind, particularly when they themselves (the authors) are performing the review, and have a vested interest in demonstrating that their AI-NERD method is better than the alternative.

Overall, I am not happy with the quality of this work.

Reviewer #4 (Remarks to the Author):

I think the paper is acceptable for publication after the previous rounds of reviews. This seems a fairly straightforward case of applying ML techniques to some field, in this case XPCS, and there is nothing fundamentally wrong with the approach. That said, getting the networks to work for given input data and desired outputs is a known challenge that requires expertise in both computation and domain science. As such the work definitely warrants publication. Without repeating the analysis or probing in depth I couldn't find anything fundamentally wrong.

I do agree with Reviewer #3's second point 'B' that it is fair to disclose who were "the experts with decades of experience". This does not detract from the paper, and of course one would ask experts one knows. Perhaps this can be added as a simple sentence along the lines of what was written in the reply to referees.

REVIEWERS' COMMENTS

Reviewer #3 (Remarks to the Author):

A: I feel that every time I examine a result in this paper closely, I spot an error. This causes me significant concern, because I did not study every single result in the paper so closely, so I believe there might well be other problems lurking elsewhere. In particular:

A1: In my previous comment 3b, I pointed out that the authors' claims about Supplemental Figure 7 were incorrect. They respond by claiming that this was just a mistake in deciding whether to use 50% or 70% accuracy thresholds, but their original claim accuracy over the first 200 suggested images does not hold at either 50% or 70%. We can see that they have modified their claim now to "50% accuracy for 260 out of the 300 suggested images".

Our original claim of 50/70% accuracy over 200 images was an honest mistake for which we apologize, and have corrected this in the previous revision, as mentioned by the reviewer. The accuracy metrics shown in Supplemental Figure 7 are presented as a clear and simple comparison to traditional analysis by humans, however we note that all judgements of similarity between C2 images is subjective by nature – this is why there was confusion between whether 50% or 70% was an appropriate accuracy threshold, and indeed why we suggest the use of automated methods for comparison.

A2: In my previous comment 3d, I again questioned the authors' claims about what is shown in Supplemental Figure 8, and they have now responded by deleting the whole figure.

We appreciated the reviewer's previous comment regarding Supplemental Figure 8. The high-dimensionality of both raw data and the latent embedding make direct quantitative comparison extremely difficult, and this difficulty is exacerbated by the fact that no all-encompassing method for comparing and analyzing non-equilibrium XPCS data exists. We agreed with the reviewer's feedback and removed the figure to keep the discussion as streamlined as possible. We thank the reviewer for their discerning eye and careful reading of the manuscript.

B: I also have significant concerns about the response to my comment 3c, in which I asked about the composition of the group of experts who performed "blind" assessments. Specifically:

B1: The authors provide a detailed justification for why we might need expert review; this is completely irrelevant, I am not arguing that expert review is never needed, I am simply requesting that the readers be provided with appropriate information about the review process.

In the previous comments from reviewers it was noted that the evaluation process was not entirely clear. Therefore, we added a description of the process and highlighted the need for accuracy evaluation as an aid to readers.

B2: The authors acknowledge in their response that the experts providing blind assessments include the authors. However, they do not include that detail in the paper. Why are the authors concealing this fact in the publication? Is it because readers will recognise it as a conflict of interest?

We are happy to include this detail in the manuscript and have modified it accordingly. We do not believe that there is a conflict of interest. We assert that the use of group members or collaborators for accuracy evaluation is common and crucial for evaluating scientific machine learning applications, where both specific domain knowledge and understanding of the task is critical.

Changes to the manuscript:

“Three authors of this paper (Q.Z., S.N., E.D.), who are experts in XPCS data collection and analysis but were not involved in ML model development or training, were asked to evaluate where the suggested C_2 were in fact similar to the target in a blind review process: each expert was handed two sets of 300 C_2 , without knowing how they were obtained or labeled.” Details of the blind test are included in the Methods section.

B3: The authors still do not provide any details about how they ensure that the review is truly blind, particularly when they themselves (the authors) are performing the review, and have a vested interest in demonstrating that their AI-NERD method is better than the alternative.

We understand how this could be unclear and have revised the manuscript accordingly (see above). The blind review was carried out as follows:

1. A target C_2 pattern was chosen with specific fringe features of interest.
2. The 300 nearest neighbors to the target image were identified using both AI-NERD and Euclidean distance in the raw data space.
3. Data for each case were provided as an unlabeled PowerPoint slide deck to domain experts (Q.Z., S.N., E.D.), who then went through all 600 C_2 suggestions and labeled them as ‘similar’ or ‘not similar’.

We emphasize the researcher who collected and prepared the data did not participate in the evaluation of the suggestions, and that the sets of C_2 images were unlabeled

such that it was impossible to tell whether suggestions came from AI-NERD or conventional analysis.

Overall, I am not happy with the quality of this work.

Reviewer #4 (Remarks to the Author):

I think the paper is acceptable for publication after the previous rounds of reviews. This seems a fairly straightforward case of applying ML techniques to some field, in this case XPCS, and there is nothing fundamentally wrong with the approach. That said, getting the networks to work for given input data and desired outputs is a known challenge that requires expertise in both computation and domain science. As such the work definitely warrants publication. Without repeating the analysis or probing in depth I couldn't find anything fundamentally wrong.

We thank the reviewer for the comments and assessment. We agree that our point is to suggest a method for combining physical domain expertise with ML methods to aid scientific research.

I do agree with Reviewer #3's second point 'B' that it is fair to disclose who were "the experts with decades of experience". This does not detract from the paper, and of course one would ask experts one knows. Perhaps this can be added as a simple sentence along the lines of what was written in the reply to referees.

We agree that having co-author's review the quality of AI-NERD suggestions does not detract from the work and are happy to include this information in the manuscript. We have modified the manuscript as follows:

"Three authors of this paper (Q.Z., S.N., E.D.), who are experts in XPCS data collection and analysis but were not involved in ML model development or training, were asked to evaluate where the suggested C_2 were in fact similar to the target in a blind review process: each expert was handed two sets of 300 C_2 , without knowing how they were obtained or labeled." Details of the blind test are included in the Methods section.